# A novel pH-dependent membrane peptide that binds to EphA2 and inhibits cell migration

Daiane S Alves[1], Justin M Westerfield[1], Xiaojun Shi[2,3,4,5], Vanessa P Nguyen[1], Katherine M Stefanski[6], Kristen R Booth[1], Soyeon Kim[2], Jennifer Morrell-Falvey[1,7], Bing-Cheng Wang[3,4,5], Steven M Abel[8,9], Adam W Smith[2], Francisco N Barrera[1]*

[1]Department of Biochemistry & Cellular and Molecular Biology, University of Tennessee, Knoxville, United States; [2]Department of Chemistry, University of Akron, Akron, United States; [3]Department of Physiology and Biophysics, Case Western Reserve University, Cleveland, United States; [4]Pharmacology, Case Western Reserve University, Cleveland, United States; [5]Rammelkamp Center for Research, MetroHealth Medical Center, Cleveland, United States; [6]Graduate School of Genome Science and Technology, University of Tennessee, Knoxville, United States; [7]Biosciences Division, Oak Ridge National Laboratory, Oak Ridge, United States; [8]Department of Chemical and Biomolecular Engineering, University of Tennessee, Knoxville, United States; [9]National Institute for Mathematical and Biological Synthesis, University of Tennessee, Knoxville, United States

**Abstract** Misregulation of the signaling axis formed by the receptor tyrosine kinase (RTK) EphA2 and its ligand, ephrinA1, causes aberrant cell-cell contacts that contribute to metastasis. Solid tumors are characterized by an acidic extracellular medium. We intend to take advantage of this tumor feature to design new molecules that specifically target tumors. We created a novel pH-dependent transmembrane peptide, TYPE7, by altering the sequence of the transmembrane domain of EphA2. TYPE7 is highly soluble and interacts with the surface of lipid membranes at neutral pH, while acidity triggers transmembrane insertion. TYPE7 binds to endogenous EphA2 and reduces Akt phosphorylation and cell migration as effectively as ephrinA1. Interestingly, we found large differences in juxtamembrane tyrosine phosphorylation and the extent of EphA2 clustering when comparing TYPE7 with activation by ephrinA1. This work shows that it is possible to design new pH-triggered membrane peptides to activate RTK and gain insights on its activation mechanism.
DOI: https://doi.org/10.7554/eLife.36645.001

*For correspondence:
fbarrera@utk.edu

Competing interests: The authors declare that no competing interests exist.

## Introduction

Eph receptors are the largest sub-group of the transmembrane receptor tyrosine kinase (RTK) family (*Kania and Klein, 2016*; *Lemmon and Schlessinger, 2010*) and are divided in two classes, EphA and EphB. Humans have nine different EphA and five EphB receptors that are activated, with some exceptions, by ephrinA and ephrinB ligands, respectively (*Lisabeth et al., 2013*). In general, Eph receptors and ephrin ligands are found on opposing cells, where they establish cell-to-cell contacts (*Kania and Klein, 2016*; *Lisabeth et al., 2013*). Full activation of Eph receptors is achieved upon clustering of receptors at the plasma membrane (*Kullander and Klein, 2002*; *Barquilla and Pasquale, 2015*; *Davis et al., 1994*). EphrinA molecules are anchored to the extracellular face of the

plasma membrane by a glycosylphosphatidylinositol linkage. Binding of ephrinA ligands to EphA causes cell repulsion through activation of intracellular signaling pathways that control cytoskeletal dynamics. As a result, the EphA-ephrinA signaling axis controls contact-dependent cell communication that drives cell adhesion, migration, morphology, and survival (*Kania and Klein, 2016*; *Boyd et al., 2014*). These activities are important during development, particularly in nervous system formation and blood vessel remodeling, and in adult homeostasis of neural, bone and epithelial tissues (*Barquilla and Pasquale, 2015*).

Not surprisingly, misregulation of EphA-ephrinA signaling can lead to pathological states. For example, it has been found that altered localization of EphA4 contributes to synaptic dysfunction in Alzheimer's disease (*Rosenberger et al., 2014*), while a missense mutation in EphA2 can cause age-related cortical cataracts (*Barquilla and Pasquale, 2015*; *Jun et al., 2009*). Moreover, Eph receptors can contribute to cancer malignancy. Indeed, Eph receptors were named after their discovery in an erythropoietin-producing hepatoma cell line (*Hirai et al., 1987*). Relevant to cancer, the EphA2-ephrinA1 signaling axis regulates events crucial for cellular transformation and malignancy. Furthermore, EphA2 is overexpressed in multiple cancer types (breast, brain, ovary, bladder, prostate, pancreas, esophagus, lung, and stomach) (*Tandon et al., 2011*). However, regulation of EphA2 is complex, and several factors, including ligand binding and downstream events, can cause EphA2 to act as a tumor suppressor or as an oncogenic protein (*Shi and Wang, 2018*).

EphA2 contains a N-terminal extracellular domain (ECD) connected to the intracellular domain (ICD) by a single transmembrane (TM) helix. Information from ephrinA1 binding to the extracellular ligand-binding domain is transmitted across the membrane by the TM helix in the form of a conformational change in the ICD. As a result, the intracellular kinase domain is activated and auto-phosphorylates multiple tyrosine residues, which triggers a signaling cascade (*Binns et al., 2000*; *Locard-Paulet et al., 2016*). Full activation of EphA2 requires first dimerization, a process mediated by the TM domain (*Bocharov et al., 2010*; *Sharonov et al., 2014*), as well as soluble domains (*Shi et al., 2017*; *Himanen et al., 2010*), followed by assembly into clusters. However, EphA2 activation is poorly understood, especially at the level of the conformational interplay between the TM and soluble domains. New tools are needed to interrogate the conformational rearrangements that mediate EphA2 activation.

We have recently designed the ATRAM (acidity-triggered rational membrane) peptide. ATRAM is a highly soluble synthetic peptide that is capable of pH-dependent interaction with lipid membranes: at neutral pH, ATRAM binds to the membrane surface, while a decrease in pH triggers insertion into the lipid bilayer as a TM helix (*Nguyen et al., 2015*). The pH-dependent membrane insertion of ATRAM results from the protonation of glutamic acid residues, as this event switches the polarity of the peptide from moderately to highly hydrophobic. The pH-triggered membrane insertion of ATRAM and similar peptides can be used to target cell membranes in acidic environments (*Reshetnyak et al., 2007*). Acidosis of the extracellular medium is a hallmark of aggressive tumors and results from altered cell metabolism and physiology (*Schornack and Gillies, 2003*). Tumor acidosis favors aggressiveness, metastasis and invasion (*Martinez-Outschoorn et al., 2011*). We reasoned that the strategy used to design ATRAM could be applied to conditionally solubilize the transmembrane domain of a receptor. Here, we have used this approach to transform the TM helix of the human EphA2 into an amphitropic peptide, called TYPE7 (transmembrane tyrosine kinase peptide for Eph). TYPE7 is a highly soluble peptide in aqueous solution that inserts into cellular membranes in a pH-dependent fashion. The TM state of TYPE7 interacts with EphA2 to induce receptor oligomerization and phosphorylation, which causes inhibition of cell migration. The observed mechanistic differences between EphA2 activation by ephrinA1 and TYPE7 provide new insights into the activation mechanism of EphA2.

## Results

### A pH decrease triggers the membrane insertion of TYPE7

TYPE7 is comprised of the sequence of the TM region of EphA2 and flanking residues (*Figure 1A*). We introduced five glutamic acid residues at the C-terminus and two in the TM region to enhance water solubility and confer pH-responsiveness. TYPE7 dissolved readily in buffer (*Figure 1—figure supplement 1*), and circular dichroism (CD) spectroscopy experiments showed that TYPE7 was

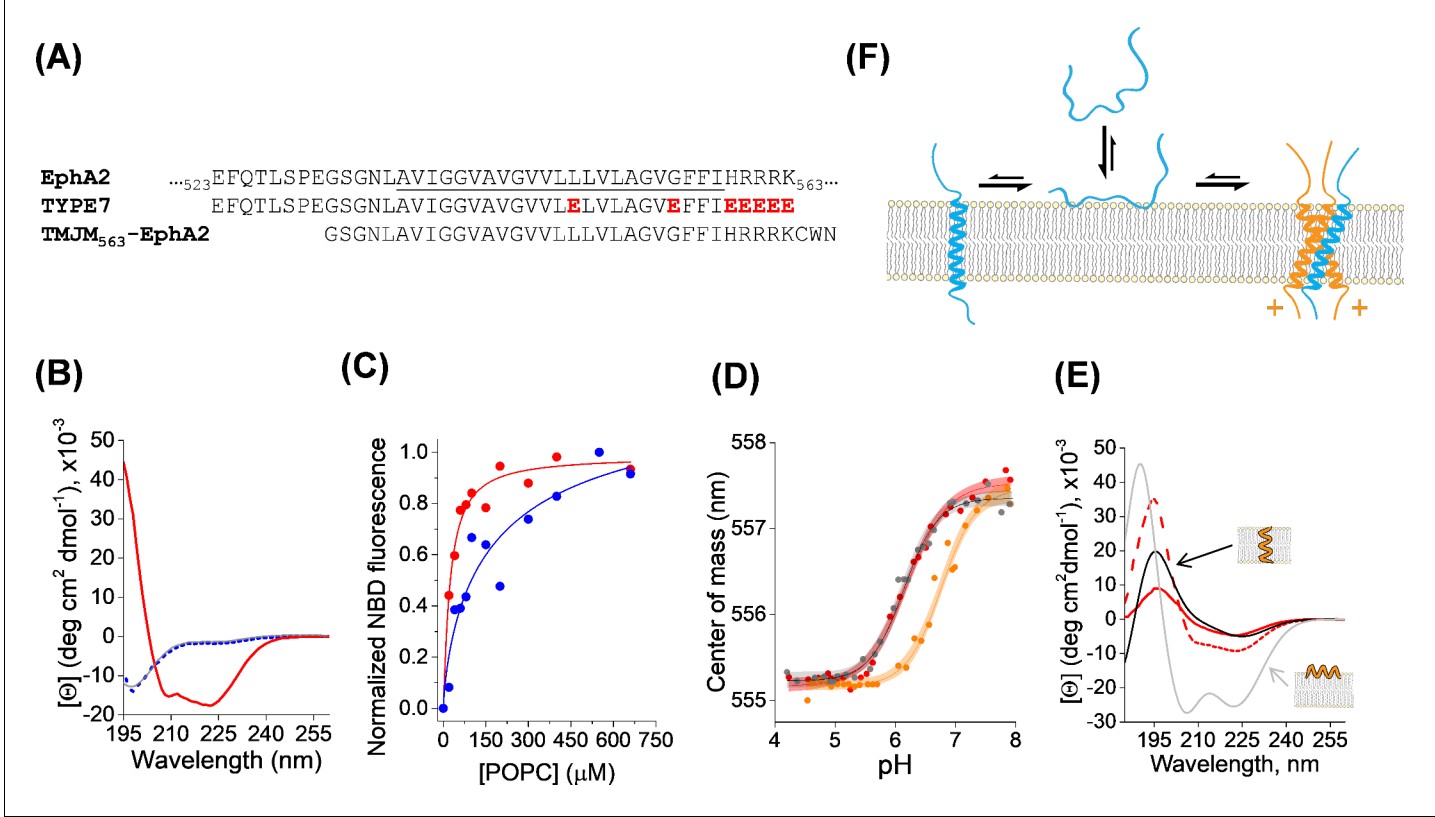

**Figure 1.** Membrane interaction of TYPE7. (**A**) *Top,* partial amino acid sequence of the human EphA2 receptor showing the TM helix (underlined), preceded by a short extracellular segment, and followed by the start of the juxtamembrane segment. Residue numbers in the sequence of EphA2 are shown. *Middle,* sequence of the TYPE7 peptide, where the acidic residues introduced are shown in red. *Bottom,* sequence of the TMJM$_{563}$-EphA2 peptide used in panel D. (**B**) Circular dichroism determination of TYPE7 secondary structure in buffer at pH 8 (grey line), and in the presence of POPC vesicles at pH 8 (dotted blue line) and after acidification to pH 4 (red line). (**C**) TYPE7 binding to POPC vesicles at pH 5 (red) and pH 8 (blue). Lines are fittings to *Equation. 3*, used to determine the *Kp* values. Lipid binding was measured using the environmentally-sensitive dye NBD attached to the N$_t$ of TYPE7. (**D**) Determination of the pH midpoint (pH$_{50}$) for the insertion of TYPE7 into POPC vesicles. TYPE7 data is shown in red symbols. Data obtained in vesicles containing the GWALP23 peptide control are shown in grey, and in vesicles containing TMJM$_{563}$-EphA2 in orange. Peptide insertion was monitored by following changes in the NBD spectral center of mass (*Equation. 1*) (*Scott et al., 2017*; *Barrera et al., 2002*). Control OCD experiments showed that TMJM$_{563}$-EphA2 formed a TM helix (*Figure 1—figure supplement 4*). The lines correspond to the fitting to the data using *Equation. 2* and 95% confidence intervals are shown as shaded areas (*n* = 6). (**E**) OCD determination of the membrane orientation of TYPE7. Data were obtained in POPC (16:0,18:1-PC, dashed red line) and 22:1,22:1-PC (continuous red line). The theoretical spectra for a perfectly transmembrane (0°, black line) and peripheral (90°, grey line) helix are shown as a reference. (**F**) Cartoon of the different states TYPE7 (blue) adopts, and how TMJM$_{563}$-EphA2 (orange) promotes the TM state of TYPE7. Arrows represent approximate equilibrium conditions found at pH ~6.5. The (+) symbols represent basic residues in the juxtamembrane segment of EphA2.

DOI: https://doi.org/10.7554/eLife.36645.002

The following figure supplements are available for figure 1:

**Figure supplement 1.** TYPE7 solubility.

DOI: https://doi.org/10.7554/eLife.36645.003

**Figure supplement 2.** TYPE7 does not induce significant membrane leakage.

DOI: https://doi.org/10.7554/eLife.36645.004

**Figure supplement 3.** TYPE7 is not toxic, and shows a pH-dependent interaction with cells.

DOI: https://doi.org/10.7554/eLife.36645.005

**Figure supplement 4.** TM-EphA2 peptide inserts into membranes as a transmembrane helix.

DOI: https://doi.org/10.7554/eLife.36645.006

unstructured in solution at neutral and slightly basic pH (as indicated by the single minimum at ~200 nm, grey line in *Figure 1B* and *Figure 1—figure supplement 1*). However, in the presence of phosphatidylcholine (POPC) lipid vesicles, TYPE7 bound to lipids (*Figure 1B,C*), without causing bilayer disruption (*Figure 1—figure supplement 2*). We used a NBD dye as a reporter for lipid interaction,

and observed that the TYPE7-NBD lipid affinity was pH-dependent. While the lipid partition coefficient ($Kp$) at pH 8 was $0.8 \times 10^6$ ($\pm 0.4 \times 10^6$), at pH 5 it increased to $2.9 \times 10^6$ ($\pm 0.4 \times 10^6$) (mean $\pm$ S.E.M, $n = 3$). This result is in agreement with our expectation of TYPE7 being more hydrophobic at acidic pH, as a result of side chain protonation of glutamic acids. Next, we performed a complete pH titration in the presence of POPC vesicles, and observed that TYPE7-NBD fluorescence changed in a sigmoidal fashion (*Figure 1D*, red line), with a pH midpoint ($pH_{50}$) of 6.18. We used CD to determine the conformation that TYPE7 adopts in the presence of lipids at neutral and acidic pH. At close to neutral pH TYPE7 was unfolded (see *Figure 1B*, dotted blue line), while at acidic pH the two characteristic α-helical minima were observed. This change indicates that the pH titration involves membrane helical formation.

Oriented circular dichroism (OCD) can determine the alignment of an α-helix with respect to the plane of hydrated supported bilayers. *Figure 1E* depicts the theoretical OCD spectra corresponding to a TM α-helix lying on the membrane surface (grey line) and inserted into the membrane (black line), where the 208 nm helical minimum is almost absent (*Wu et al., 1990*; *Bürck et al., 2016*; *Ulmschneider et al., 2014*). We used OCD to determine the helical membrane orientation that TYPE7 adopts in POPC at pH 5. We observed that the OCD spectrum (*Figure 1E*, dashed red line) was closer to the theoretical curve for a TM helix. However, the differences with the black line suggest that TYPE7 adopted a tilted TM helix orientation in POPC bilayers. Transmembrane peptides will typically tilt in the membrane to avoid hydrophobic mismatch (*Killian and Nyholm, 2006*). We reasoned that if TYPE7 formed a TM helix, its membrane tilt would decrease to adapt to a thicker lipid membrane (*Anbazhagan and Schneider, 2010*). To test this idea, we repeated the OCD experiment in 22:1,22:1-PC, a lipid with longer acyl chains that forms bilayers 7.3 Å thicker than POPC (16:0,18:1-PC) (*Kucerka et al., 1808*; *Kucerka et al., 2009*). In agreement with the expected behavior for a TM helix, the OCD spectrum in the thicker bilayer was closer to the theoretical TM curve with no tilt. Taken together, our data reveal that the sigmoidal pH titration (*Figure 1D*), associated with increased lipid affinity (*Figure 1C*), represents the transition from an unstructured state bound to the membrane surface to a transmembrane helix found at lower pH (*Figure 1F*).

## TYPE7 shows no toxicity and binds to cells in a pH-dependent manner

We explored if TYPE7 was also able to bind to cellular membranes. To this end, we studied the cellular interaction of a fluorescently labeled version of TYPE7 at different pH values (*Figure 1—figure supplement 3A*). We observed a robust interaction with cells at neutral pH, which increased with acidification. This indicates that the enhanced lipid affinity at acidic pH values is observed both in lipid vesicles and in cells. However, since satisfactory TYPE7 cell binding was achieved at neutral pH, we decided to employ physiological pH for the ensuing cellular experiments. Additionally, we performed cell viability experiments to study if TYPE7 was toxic to cells. The results of an MTS assay indicated that the peptide did not decrease cell viability (*Figure 1—figure supplement 3B*).

## TYPE7 interacts with EphA2

Next, we investigated if TYPE7 could interact with EphA2. The single TM helix of EphA2 forms a dimer that mediates receptor dimerization (*Bocharov et al., 2010*; *Sharonov et al., 2014*; *Singh et al., 2015*). The TM region of TYPE7 contains glutamic acid residues designed to align into a single helical face. NMR studies indicated that this face is located away from the dimerization interface of EphA2 (*Figure 1—figure supplement 3C*) (*Bocharov et al., 2010*). As a result, TYPE7 theoretically contains an intact dimerization interface to interact with the EphA2 TM helix. We hypothesized that this would allow binding of TYPE7 to the transmembrane domain of EphA2. To evaluate this hypothesis, we used a new peptide encompassing the TM domain of EphA2 and five residues at the N-terminus of the juxtamembrane segment (JMS), through residue K563 (*Figure 1A*). We refer to this peptide as $TMJM_{563}$-EphA2 (*Figure 1—figure supplement 4*). We used the $pH_{50}$ assay to study the interaction between TYPE7 and $TMJM_{563}$-EphA2. We reasoned that transmembrane binding between $TMJM_{563}$-EphA2 and TYPE7 would increase the $pH_{50}$, as the TM state of TYPE7 would be stabilized over the surface-bound conformation, displacing the equilibrium (*Figure 1F*). Indeed, the presence in the vesicles of $TMJM_{563}$-EphA2 at a 4-fold molar excess, increased the $pH_{50}$ of TYPE7 from $6.18 \pm 0.09$ to $6.85 \pm 0.16$ (mean $\pm$S.D., $n = 7 - 9$). Interestingly, *Figure 1D* shows that, in these conditions, a non-negligible fraction of TYPE7 is already in the TM

state at pH 7. This suggests the intriguing possibility that TYPE7 could interact with EphA2 without requiring strong acidification. To study the specificity of this effect we performed a control experiment replacing TMJM$_{563}$-EphA2 by GWALP23, an unrelated peptide that also forms a transmembrane helix (*Ozdirekcan et al., 2005*; *Holt et al., 2009*; *Rankenberg et al., 2012*). The pH$_{50}$ of TYPE7 was similar in the absence or presence of GWALP23 (6.18 ± 0.09 and 6.17 ± 0.20, respectively, *Figure 1D*), suggesting that TMJM$_{563}$-EphA2 specifically interacts with TYPE7.

To explore the cellular relevance of the biophysical results, we examined the co-localization of TYPE7 with endogenous EphA2 in A375 cells at physiological pH. We evaluated the effect of EphA2 activation on the interaction between TYPE7 and EphA2 by treating the cells with ephrinA1-Fc (EA1). EA1 uses a Fc group (heavy chain of human IgG1) to crosslink ephrinA1. Incubation with EA1 recapitulates EphA2 trans-activation by membrane clusters of ephrinA1 (*Davis et al., 1994*). The resulting EphA2 clustering and phosphorylation leads to recycling into endosomes and degradation (*Kania and Klein, 2016*; *Nikolov et al., 2014*; *Boissier et al., 2013*; *Sabet et al., 2015*). We used confocal microscopy to study the cellular distribution of EphA2. As expected, we observed that EA1 promoted EphA2 clustering, resulting in accumulation of large puncta at the plasma membrane and cytosolic recycling (compare two insets in left column of *Figure 2A*). We used TYPE7 fluorescently labelled with Alexa568 to assess co-localization with EphA2. We observed that the TYPE7 signal overlapped to a large degree with the EphA2 receptor in the plasma membrane (*Figure 2A*, upper right panel). However, this could partially result from the membrane affinity of TYPE7 (*Figure 1C*). To test the specificity of the co-localization, we performed additional experiments in the presence of EA1, and evaluated if TYPE7 partitioned to the clusters. Interestingly, after EA1 activation we observed stronger TYPE7 co-localization with EphA2 (*Figure 2A*, lower right panel). We quantified co-localization using the Pearson correlation coefficient (*r*) (*Figure 2B*) (*Manders et al., 1992*), which showed that the positive pixel correlation between EphA2 and TYPE7 (*r* = 0.26, *n* = 14) increased significantly upon receptor activation with EA1 (*r* = 0.38, *n* = 17) (*t* = −2.68, p<0.05). Next, we performed a co-precipitation assay to confirm the interaction between TYPE7 and endogenous EphA2. To this end, we treated H358 cells with TYPE7 labelled with a near-IR fluorophore, DyLight 680 (TYPE7-DL). After EphA2 immuno-precipitation, SDS-PAGE gels showed a fluorescent band corresponding to the molecular weight of TYPE7-DL (5.2 KDa) (*Figure 2C*). Interestingly, when EphA2 was activated with EA1, the amount of TYPE7 that precipitated with endogenous EphA2 increased fourfold. These data suggest that the peptide might be trapped in EphA2 clusters. Taken together, the co-localization and co-precipitation results indicate that TYPE7 interacts with EphA2 in cells, and binding is enhanced upon activation of EphA2.

## TYPE7 inhibits cell migration by specific EphA2 phosphorylation at Y772 and decreases Akt phosphorylation

Next, we determined the functional significance of TYPE7 binding to EphA2. EphA2 controls cell-cell contact, and EphA2 activation inhibits cell migration. The effect of TYPE7 on EphA2-mediated cell migration was tested by using a Boyden chamber assay. EA1 was used as a positive control of ligand-induced inhibition of cell migration (*Miao et al., 2000*). *Figure 3A* shows that incubation with TYPE7 reduced A375 cell migration to a similar degree as EA1. Co-incubation of TYPE7 with EA1 did not further inhibit cell migration, indicating that a maximum inhibitory effect had already been obtained with saturating levels of EA1. When we repeated the Boyden chamber assay in H358 cells, we observed that TYPE7 also efficiently inhibited cell migration in this cell type (*Figure 3—figure supplement 1*).

Activation of EphA2 by EA1 causes phosphorylation of tyrosine residues in the juxtamembrane segment (JMS) of the ICD (Y588 and Y594) and the kinase domain activation loop (Y772). Phosphorylation of these residues is followed by a signaling cascade that inhibits cell migration and invasion (*Locard-Paulet et al., 2016*). To understand TYPE7 anti-migratory effects, we performed Western blots using EphA2 phospho-specific antibodies in H358 cells. We found that incubation with TYPE7 increased phosphorylation of Y772 as efficiently as EA1 (*Figure 3B,C*). Y772 is located in the activation loop of the kinase domain (*Fang et al., 2008*; *Balasubramaniam et al., 2011*), and phosphorylation at this site is critical for ligand-dependent inhibition of trans-endothelial migration controlled by EphA2 (*Locard-Paulet et al., 2016*). To evaluate the specificity of the action of TYPE7 on EphA2, we performed control experiments with pHLIP. This peptide displays a similar pH-dependent membrane insertion to TYPE7 (*Hunt et al., 1997*; *Scott et al., 2017*), and has a similar content of acidic

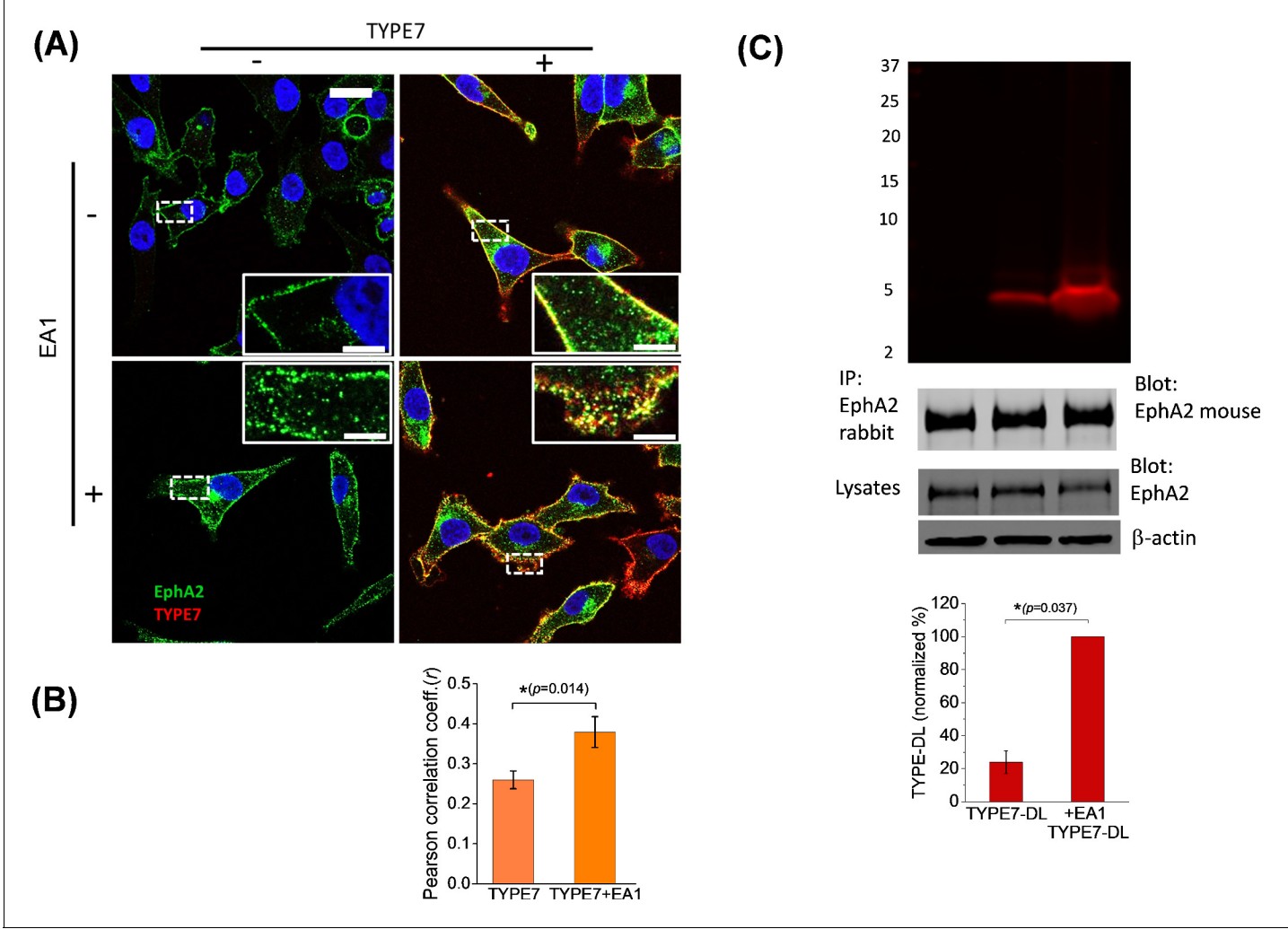

**Figure 2.** TYPE7 interacts with endogenous EphA2 in cells. (**A**) Confocal microscopy shows co-localization of TYPE7 and EphA2. A375 cells were incubated in the presence (+) or absence (-) of 0.5 µg/mL EA1 and 0.2 µM TYPE7-Alexa568 (red) for 5 min at room temperature. Cells were fixed and endogenous EphA2 was labeled via immunofluorescence (green). Images were collected using a 63x objective, and insets show images corresponding to the white dashed areas collected with a 100x objective. Scale bars are 20 µm and 5 µm, respectively. (**B**) The Pearson correlation coefficient (r) was calculated for cells incubated with TYPE7 in the absence and presence of EA1. Bar graph shows mean ±S.D. Student's *t*-test was performed for 14 – 17 images. *p<0.05, with as effect size of 0.80 standard deviations, *n* = 2. (**C**) *Top*, SDS-PAGE showing that TYPE7-DL co-precipitates with endogenous EphA2 when using a polyclonal anti-rabbit EphA2 antibody. *Middle*, control Western blots of EphA2 immunoprecipitation blotted with mouse anti-EphA2 show that similar amounts of endogenous EphA2 were pulled down in all samples. Total cell lysates blotted with EphA2 and β-actin indicate that similar levels of protein were loaded. *Bottom*, quantification of the fluorescent bands. Bar graph shows mean ±S.D. as a percentage of maximum intensity. A Mann-Whitney test was performed (*p<0.05), *n* = 3.

DOI: https://doi.org/10.7554/eLife.36645.007

residues (*Barrera et al., 2011*; *Fendos et al., 2013*), but pHLIP displays low sequence homology with TYPE7 (*Figure 3—figure supplement 2A*). Specifically, we evaluated if EphA2 phosphorylation at Y772 or cell migration were affected by the membrane insertion of pHLIP. We observed that the presence of pHLIP changed neither EphA2 Y772 phosphorylation (*Figure 3—figure supplement 2*) nor cell migration (*Figure 3—figure supplement 3*), suggesting that the effect of TYPE7 is specific.

Intriguingly, TYPE7 and EA1 caused different JMS phosphorylation, as TYPE7 did not promote phosphorylation of Y588 and Y594 (*Figure 3B,D,E*). Additionally, we observed that TYPE7 did not induce cell proliferation or phosphorylation of S897 (*Figure 3—figure supplement 4A–C*), a residue phosphorylated by Akt, RSK, and PKA that promotes ligand-independent cell migration and invasion (*Miao et al., 2009*; *Wang, 2011*; *Zhou et al., 2015*; *Barquilla et al., 2016*). TYPE7 did not cause

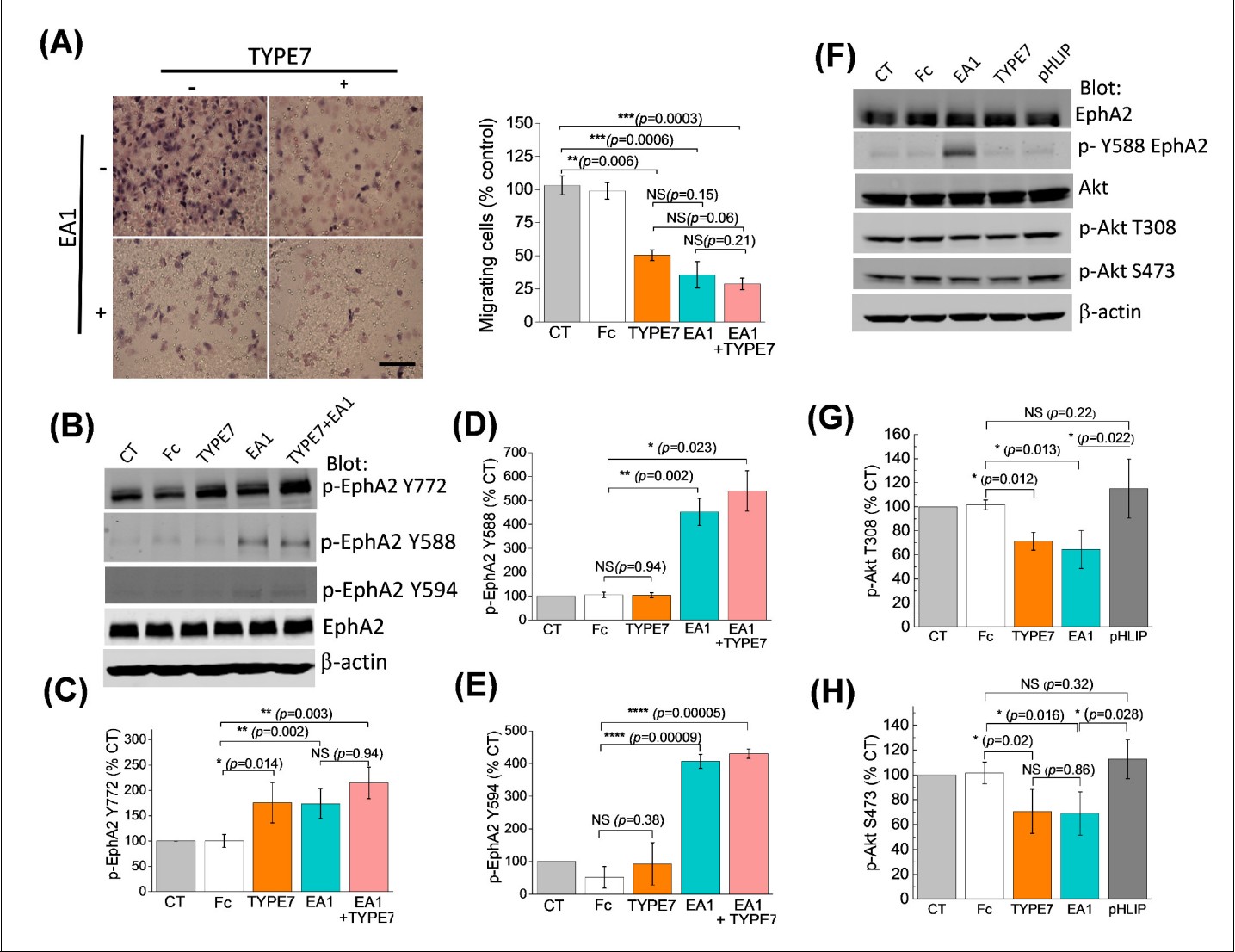

**Figure 3.** TYPE7 decreases cell migration, and induces EphA2 phosphorylation at Y772 and Akt de-phosphorylation. (**A**) *Left*, cell migration was measured in the presence and absence of TYPE7 and EA1 using a Boyden cell chamber assay. Representative images are shown. *Right*, quantification of migrating cells, showing that incubation with TYPE7 reduced A375 cell migration to a similar degree as EA1, with effect sizes of 8.4 and 12.6 standard deviations from control, respectively. $N = 3$. Cells were treated with an isolated Fc group as a control for the Fc present in EA1. Scale bar is 200 μm (**B–E**), Phosphorylation of Y772 and JMS phosphorylation at Y588 and Y594. A representative Western blot is shown (**B**). Band intensity was quantified for p-Y772 (**C**), p-Y588 (**D**), and p-Y594 (**E**). We found that incubation with TYPE7 increased phosphorylation of Y772 as efficiently as EA1, with effect sizes of 5.1 and 7.7 standard deviations from control, respectively. Mean ±S.D. are shown. $n = 5$. (**F–H**), Phosphorylation levels of Akt. A representative Western blot is shown (**F**) and band intensity was quantified for p-T308 (**G**) and p-S473 (**H**). Lysates were blotted against total EphA2 to detect total protein levels, and β-actin as a loading control. Student's *t*-test was performed to obtain *p* values (*p<0.05; **p<0.01; ***p<0.001; ****p<0.0001 and NS, not significant).

DOI: https://doi.org/10.7554/eLife.36645.008

The following figure supplements are available for figure 3:

**Figure supplement 1.** TYPE7 decreases cell migration in H358 cells.
DOI: https://doi.org/10.7554/eLife.36645.009

**Figure supplement 2.** The control pHLIP peptide does not affect the phosphorylation of EphA2 at Y772.
DOI: https://doi.org/10.7554/eLife.36645.010

**Figure supplement 3.** The control pHLIP peptide does not affect cell migration.
DOI: https://doi.org/10.7554/eLife.36645.011

**Figure supplement 4.** EphA2 expression levels and phosphorylation at S897 are not affected by TYPE7.
DOI: https://doi.org/10.7554/eLife.36645.012

*Figure 3 continued on next page*

*Figure 3 continued*

**Figure supplement 5.** Human Phospho-Tyrosine RTK array.

DOI: https://doi.org/10.7554/eLife.36645.013

EphA2 expression changes either (*Figure 3—figure supplement 4D–E*). Last, we examined the specificity of TYPE7 using an array of 49 human RTK. The array data suggests that TYPE7 does not significantly increase tyrosine phosphorylation of other RTKs (*Figure 3—figure supplement 5*). Taken together, these results suggest that TYPE7 inhibits cell migration by inducing specific EphA2 phosphorylation at Y772, but not at the JMS.

The activation of EphA2 by ephrins elicits downstream signaling that inhibits cell migration (*Shi and Wang, 2018*). Akt is an important downstream target of EphA2 (49). When Akt is activated, it is phosphorylated at residues T308 and S473. Activation of EphA2 by ephrinA1 inhibits Akt and reduces phosphorylation at the two sites (*Barquilla et al., 2016*). We evaluated the effect of TYPE7 on the phosphorylation of Akt. We observed that TYPE7 significantly reduced phosphorylation at T308 and S473 to a degree that is comparable to the effect of EA1 (*Figure 3F–H*). Again, no changes were observed when pHLIP was used as a negative control.

## TYPE7 promotes limited self-assembly of EphA2

In the absence of ligand, EphA2 is found in a monomer-dimer equilibrium (*Singh et al., 2015*). However, differently to other receptor tyrosine kinases, EphA2 dimerization does not cause full receptor activation (*Nikolov et al., 2014*; *Janes et al., 2012*). Instead, stronger EphA2 activation is achieved upon dimer self-assembly into higher-order clusters that form extended signaling arrays. These clusters can contain hundreds of EphA2 molecules (*Himanen et al., 2010*; *Nikolov et al., 2014*; *Janes et al., 2012*) and appear as micron-sized puncta in the plasma membrane (*Salaita et al., 2010*). We explored if TYPE7 activates EphA2 by promoting receptor clustering. First, we employed super-resolution Structured Illumination Microscopy (SIM) to qualitatively investigate this possibility. *Figure 4A* shows that untreated cells have a relatively homogeneous EphA2 distribution at the plasma membrane. EA1 treatment caused EphA2 to concentrate in brighter foci on the membrane, indicating clusters of EphA2 (marked as white arrowheads). Strikingly, incubation with TYPE7 did not promote foci formation. Similar conclusions were drawn from confocal imaging (see *Figure 2*). This was surprising, since TYPE7 increased Y772 phosphorylation and reduced cell migration as effectively as EA1, but apparently, it did so without promoting formation of EphA2 foci. This suggests that TYPE7 and EA1 might achieve similar inhibition of cell migration despite inducing different levels of EphA2 self-assembly.

To confirm these results, we used fluorescence correlation spectroscopy (FCS) to study EphA2 lateral organization in live cells. FCS is more sensitive than SIM for detecting changes in oligomerization status, particularly for small oligomers. FCS records the time-resolved fluorescence fluctuations within a confocal detection volume caused by diffusion of EphA2. By performing correlation analysis on the recorded fluctuation signals, auto-correlation function (ACF) curves are obtained (*Figure 4—figure supplement 1A–B*). From the ACF curve, we determined the lateral mobility of EphA2, reported as an effective diffusion coefficient (D). We investigated changes in EphA2 oligomeric state monitoring lateral mobility after TYPE7 and EA1 treatment. Although it is difficult to use lateral mobility to calculate the absolute size of EphA2 oligomers, there is a direct correlation between lateral mobility and oligomer size (*Shi et al., 2017*; *Chung et al., 2010*). Namely, for the same receptors in the same membrane environment, a decrease in the lateral mobility indicates growth in oligomer size. FCS measurements were recorded in live DU-145 cells (*Figure 4B*, *Figure 4—figure supplement 1D*) that stably express EphA2 labelled with enhanced GFP (EphA2FL-GFP) (*Shi et al., 2017*). While this experimental setting does not allow ruling out the presence of more than one diffusing component, a single relaxation term (*Equation 5*) fitted the data well. In untreated cells, the median D value for EphA2FL-GFP was 0.30 $\mu m^2$/s (*Figure 4C*, first column). When treated with EA1, the median D value decreased to 0.09 $\mu m^2$/s (*Figure 4C*, orange area). The decrease in D upon EA1 stimulation showed that, as expected, EphA2FL-GFP formed clusters (*Shi et al., 2017*). However, upon treatment with TYPE7, D decreased to 0.20 $\mu m^2$/s (*Figure 4C*, second column). This indicates that EphA2FL-GFP oligomerizes upon TYPE7 treatment, but the intermediate D value indicates that

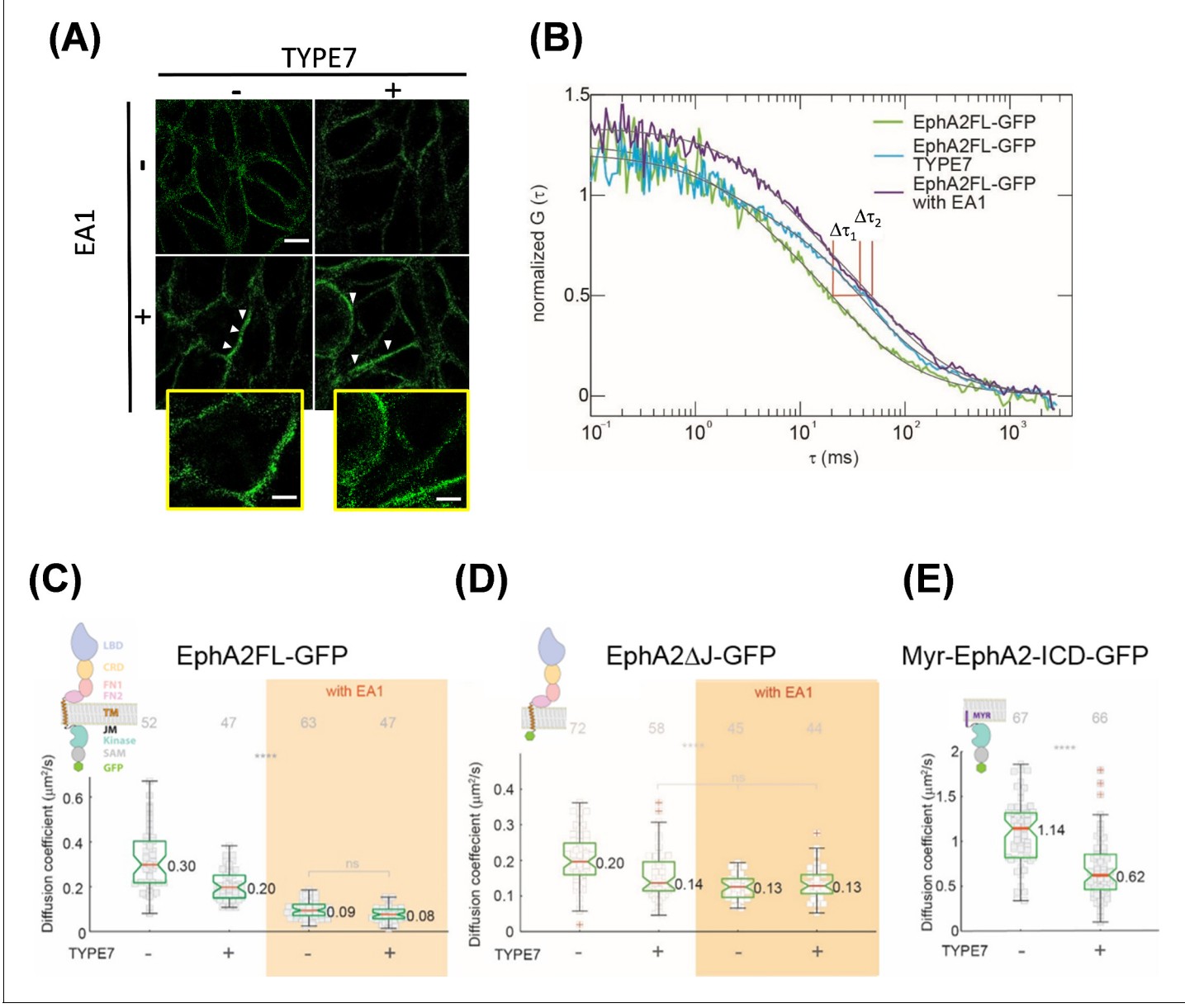

**Figure 4.** TYPE7 induces formation of oligomers of EphA2. (A) Super-resolution SIM data. H358 cells were incubated in the presence (+) or absence (-) of 0.5 μg/mL EA1 and 2 μM TYPE7. Representative images show fluorescence obtained using an anti-EphA2 antibody (*n* = 4). Scale bar is 10 μm. Insets magnify areas with clusters, and the scale bars are 5 μm. (B) Representative FCS autocorrelation curves for EphA2FL-GFP in control conditions (green) or in the presence of TYPE7 (blue) and EA1 (magenta). $\Delta\tau1$ and $\Delta\tau2$ represent the changes in dwell time. (C–E) Diffusion coefficient results, containing graphic models describing the EphA2 constructs used. (C) Box-whisker plot of measurement of the FCS diffusion coefficient of EphA2FL-GFP. (D) Diffusion coefficient of EphA2ΔJ-GFP. (E) Diffusion coefficient of Myr-EphA2 ICD-GFP. Diffusion coefficients collected from cells with and without TYPE7 treatment are reported along with EA1 ligand stimulation (orange boxes). The median values are reported next to the box plots. Each data point is the average of five 10 s FCS measurements on one cell. The grey numbers on top of the plots are the total number of cells measured. Criteria for the box, median, quartiles, whiskers and outliers are described elsewhere (*Shi et al., 2017*). One-way ANOVA tests were performed to obtain the *p* values (****$p<0.0001$; ns, not significant).

DOI: https://doi.org/10.7554/eLife.36645.014

The following figure supplements are available for figure 4:

**Figure supplement 1.** FCS supplement.
DOI: https://doi.org/10.7554/eLife.36645.015

**Figure supplement 2.** TYPE7 does not affect diffusion of PlexinA4, another single-pass transmembrane receptor.
DOI: https://doi.org/10.7554/eLife.36645.016

**Figure supplement 3.** Human phospho-kinase array studies of TYPE7 specificity.

*Figure 4 continued on next page*

*Figure 4 continued*

DOI: https://doi.org/10.7554/eLife.36645.017

EphA2 is detected in a lower-order oligomeric state than the cluster. The difference between diffusion coefficients obtained for EA1-activated EphA2FL-GFP, both with and without TYPE7, was not statistically significant. This suggests that regardless of the presence of TYPE7, EA1 caused EphA2FL-GFP to form clusters of similar size. This apparent saturation effect agrees with the cell migration and phosphorylation data (*Figure 3A,D and E*). Additionally, FCS data analysis allows us to quantify the plasma membrane levels of EphA2FL-GFP. *Figure 4—figure supplement 1C* shows that incubation with TYPE7 did not alter the levels of EphA2 expression, in agreement with Western blot data shown in *Figure 3—figure supplement 4D–E*.

To demonstrate that TYPE7 is specifically targeting EphA2 without affecting other single-pass transmembrane receptors, we tested the effect of TYPE7 on Plexin A4. Plexin A4 is a cell surface protein that has a similar domain structure as EphA2: one transmembrane domain, a large ectodomain, and an enzymatic cytoplasmic domain. Previous work showed that Plexin A4 forms an inactive dimer prior to ligand stimulation (*Marita et al., 2015*). COS-7 cells were transiently transfected with Plexin A4 labelled with eGFP (Plexin A4-eGFP). FCS measurements were carried out on the peripheral membrane area of live cells expressing Plexin A4-eGFP to measure any change in their lateral mobility upon TYPE7 treatment. In untreated cells, the median diffusion coefficient (D) value for Plexin A4-eGFP was 0.28 $\mu m^2$/s (*Figure 4—figure supplement 2*), similar to the previously published value (*Marita et al., 2015*). There was no significant difference when the cells were treated with TYPE7 (D = 0.27 $\mu m^2$/s). This control experiment suggests that TYPE7 does not affect the diffusion of transmembrane proteins in a non-specific manner.

The results obtained in lipid vesicles containing TMJM$_{563}$-EphA2 suggested that TYPE7 interacts with the membrane-proximal region of EphA2. However, TMJM$_{563}$-EphA2 encompasses not only the TM helix of EphA2 but also the first five basic JMS residues (*Figure 1A*). In order to define the domains of EphA2 that interact with TYPE7, we performed additional FCS experiments with two deletion EphA2 constructs. We first used a truncation construct where the full ICD was deleted at the first JMS residue (*Figure 4D*). The resulting construct, EphA2ΔJ-GFP (*Shi et al., 2017*), was used to study the ability of TYPE7 to target the EphA2 TM helix. Using this construct, we observed that TYPE7 treatment decreased the mobility of EphA2ΔJ-GFP from 0.20 $\mu m^2$/s to 0.14 $\mu m^2$/s, suggesting that TYPE7 binding to the TM domain increased oligomerization. Interestingly, upon EA1 stimulation, D was 0.13 $\mu m^2$/s (*Figure 4C*, orange area), similar to the value observed with TYPE7. This suggests that in the absence of the ICD, TYPE7 has a similar effect as EA1 on self-assembly, suggesting that the ICD domains might be responsible for the differences in clustering observed between EA1 and TYPE7.

Finally, we studied the oligomerization of the isolated EphA2 ICD. FCS was thus performed using Myr-EphA2 ICD-GFP transfected in COS-7 cells. In Myr-EphA2 ICD-GFP, the ICD of EphA2 is anchored to the membrane at the first JMS residue using a myristoyl group (*Shi et al., 2017*). When we performed FCS experiments with this construct, we observed faster diffusion compared to the other two EphA2 construct in control conditions. Interestingly, treatment with TYPE7 also decreased D (*Figure 4E*), indicating that TYPE7 also promoted oligomerization of the ICD (cell images are shown at *Figure 4—figure supplement 1D*). We performed control experiments to evaluate if the oligomerization change that TYPE7 induces in Myr-EphA2 ICD-GFP might result from nonspecific interactions with the myristoyl moiety. To this end, we assayed the effect of TYPE7 on six Src family kinases, which are also linked to the membrane *via* myristoylation. Such experiments showed that TYPE7 did not alter the phosphorylation status of any of the myristoylated kinases (*Figure 4—figure supplement 3*). Additionally, we assayed the phosphorylation status of 37 other protein kinases and kinase substrates, with the exception of Akt (*Figure 4—figure supplement 3C*). Importantly, we observed that TYPE7 did not induce phosphorylation changes in any of these proteins. These results additionally suggest that the effects of TYPE7 on cell migration results from changes in EphA2 activity, and not any of these other cellular targets (*Figure 4—figure supplement 3*). Collectively, our data indicate that TYPE7 interacts with both the TM helix and the ICD of EphA2, to promote receptor oligomerization.

## Discussion

In this work, we show how strategic addition of acidic residues can transform a transmembrane domain into a water-soluble species, which can be triggered to insert into membranes. This finding can have important implications for the design of new ligands that modulate protein-protein interactions in membrane proteins. A molecule capable of establishing protein-protein interactions efficiently in cellular membranes should have three fundamental properties (*Stone and Deber, 2017*): (1) be easily deliverable into the membrane, where it should reside stably; (2) adopt an appropriate conformation to bind to the target; and (3) do not cause membrane disruption. Our data indicate that TYPE7 satisfies all these criteria. TYPE7 displays affinity for lipid bilayers, while it is readily soluble in buffer, which allows for easy plasma membrane delivery in physiological conditions. We hypothesize membrane binding of TYPE7 is initially driven by its moderately hydrophobic nature, as in the ATRAM and pHLIP peptides (*Reshetnyak et al., 2007*; *Hunt et al., 1997*; *Deacon et al., 2015*). After localizing at the surface of lipid vesicles, TYPE7 adopts a TM configuration, triggered by a pH decrease.

Our studies in cells, including co-precipitation (*Figure 2*), indicated that TYPE7 interacts with endogenous EphA2. Additional experiments in a reconstituted vesicle system showed that the acidity required for TYPE7 insertion significantly diminished in the presence of the membrane region of EphA2. In fact, in the presence of $TMJM_{563}$-EphA2, the $pH_{50}$ of TYPE7 membrane insertion shifted to a less acidic value, and the transition started at neutral pH (*Figure 1D*). No changes in $pH_{50}$ were observed using a control transmembrane domain of different sequence, indicating that the interaction is specific. We propose that binding to $TMJM_{563}$-EphA2 shifts the membrane equilibrium of TYPE7 away from the membrane surface, and promotes glutamic acid protonation and formation of the transmembrane state (*Figure 1F*).

The data obtained with $TMJM_{563}$-EphA2 suggest an interaction between TYPE7 and the hydrophobic amino acids of the transmembrane helix of EphA2. However, the $TMJM_{563}$-EphA2 peptide contains at the C-terminus a basic stretch, $^{559}HRRRK^{563}$, corresponding to the start of the JMS. It has not escaped our notice that TYPE7 contains a potentially complementary acidic stretch at the C-terminus, with sequence EEEEE (*Figure 1A*), which might establish an attractive electrostatic interaction with the basic stretch of $TMJM_{563}$-EphA2. We performed additional experiments to determine if TYPE7 could interact with the JMS of EphA2 in cells. Indeed, we observed that TYPE7 promoted self-assembly of the full ICD, containing the JMS but not the TM domain, as determined by FCS (*Figure 4E*). As expected, TYPE7 also promoted self-assembly of the EphA2 construct lacking the full JMS, but containing the TM domain (*Figure 4D*). Taken together our data suggest that TYPE7 interacts with EphA2 both at the TM domain and the ICD, and we hypothesize the ICD interaction occurs at the JMS.

We studied the biological effect of the interaction of TYPE7 with EphA2 using a trans-well migration assay. Interestingly, we observed that TYPE7 inhibited EphA2-driven cell migration to a similar extent as the saturating EA1 concentrations employed (*Figure 3A*). It has been shown that phosphorylation of the activation loop residue Y772 of EphA2 is required for ligand-induced inhibition of cell migration (*Locard-Paulet et al., 2016*; *Singh et al., 2015*). To determine the molecular mechanism of the activation of EphA2 by TYPE7, we studied the phosphorylation at the JMS and kinase activation loop. We observed that EA1 and TYPE7 caused a similar increase in Y772 phosphorylation, indicating that this molecular event might explain the similar effect of both ligands on cell migration.

Surprisingly, clear differences existed in the phosphorylation of the JMS residues Y588 and Y594. While TYPE7 did not affect their state, EA1 strongly promoted phosphorylation of Y588 and Y594. JMS phosphorylation is required for EA1 activation of EphA2, since the JMS auto-inhibits the kinase domain (*Lemmon and Schlessinger, 2010*). This regulatory mechanism involves docking of the JMS to the kinase domain, which stabilizes the inactive kinase state. EphA2 activation by ephrin binding promotes phosphorylation of the JMS residues Y588 and Y594, which causes a conformational change in the JMS that leads to its release from the kinase domain, and ends auto-inhibition (*Wybenga-Groot et al., 2001*). As a result, Y772 in the kinase activation loop is phosphorylated and the kinase domain is activated (*Singh et al., 2015*; *Fang et al., 2008*). Our results show that TYPE7 promotes full EphA2 Y772 phosphorylation and inhibition of cell migration without JMS phosphorylation. This suggests that phosphorylation of the JMS is not the only path to release juxtamembrane inhibition of EphA2. How can TYPE7 release the auto-inhibition without JMS phosphorylation? We

hypothesize that the interaction between TYPE7 and the JMS of EphA2 might induce a conformational change that reorients the JMS without requiring phosphorylation, and as a result preclude autoinhibition by binding of this segment to the kinase domain. Interestingly, it has been recently reported that the regulation of the phosphorylation of Y772 and Y588 can be uncoupled by differential de-phosphorylation (*Locard-Paulet et al., 2016*). Our data indicates that phosphorylation of Y772 can occur *via* a different mechanism that does not require JMS phosphorylation. Our results illustrate the flexibility of molecular events involved in the interplay between the JMS and kinase domain, and might suggest that additional modes of release of autoinhibition could regulate EphA2 phosphorylation.

Crosstalk between Akt and EphA2 has been documented in several studies (*Shi and Wang, 2018*; *Miao et al., 2009*; *Yang et al., 2011*). Akt is a key protein that controls cell migration and differentiation through the oncogenic Akt/mTORC1 pathway (*Altomare and Khaled, 2012*). EphA2 activation by ephrinA1 downregulates this pathway through Akt de-phosphorylation mediated by a serine/threonine phosphatase (*Yang et al., 2011*). *Figure 3F–H* shows that TYPE7 decreased phosphorylation at the two main Akt kinase activation sites, T308 and S473, similarly to EA1. We propose that inhibition of Akt by TYPE7 can explain the strong inhibition of cell migration shown in *Figure 3A*. Furthermore, this observation suggests that TYPE7 can be used to inhibit the oncogenic Akt/mTORC1 signaling pathway.

EphA2 ligand-dependent activation involves formation of large clusters. We compared the effect of EA1 and TYPE7 on clustering. The FCS and SIM data in *Figure 4* show that while EA1 promotes formation of large clusters of the full-length EphA2, TYPE7 does not induce clusters, but smaller oligomers. This indicates the possibility that the large EphA2 clusters that EA1 induces are not required for EphA2-mediated inhibition of cell migration. Based on this result, we suggest that a smaller oligomer might be the active signaling state of EphA2 (*Figure 5*). A similar scenario has been proposed for EphB2 using chemical dimerizers (*Schaupp et al., 2014*). The larger EphA2

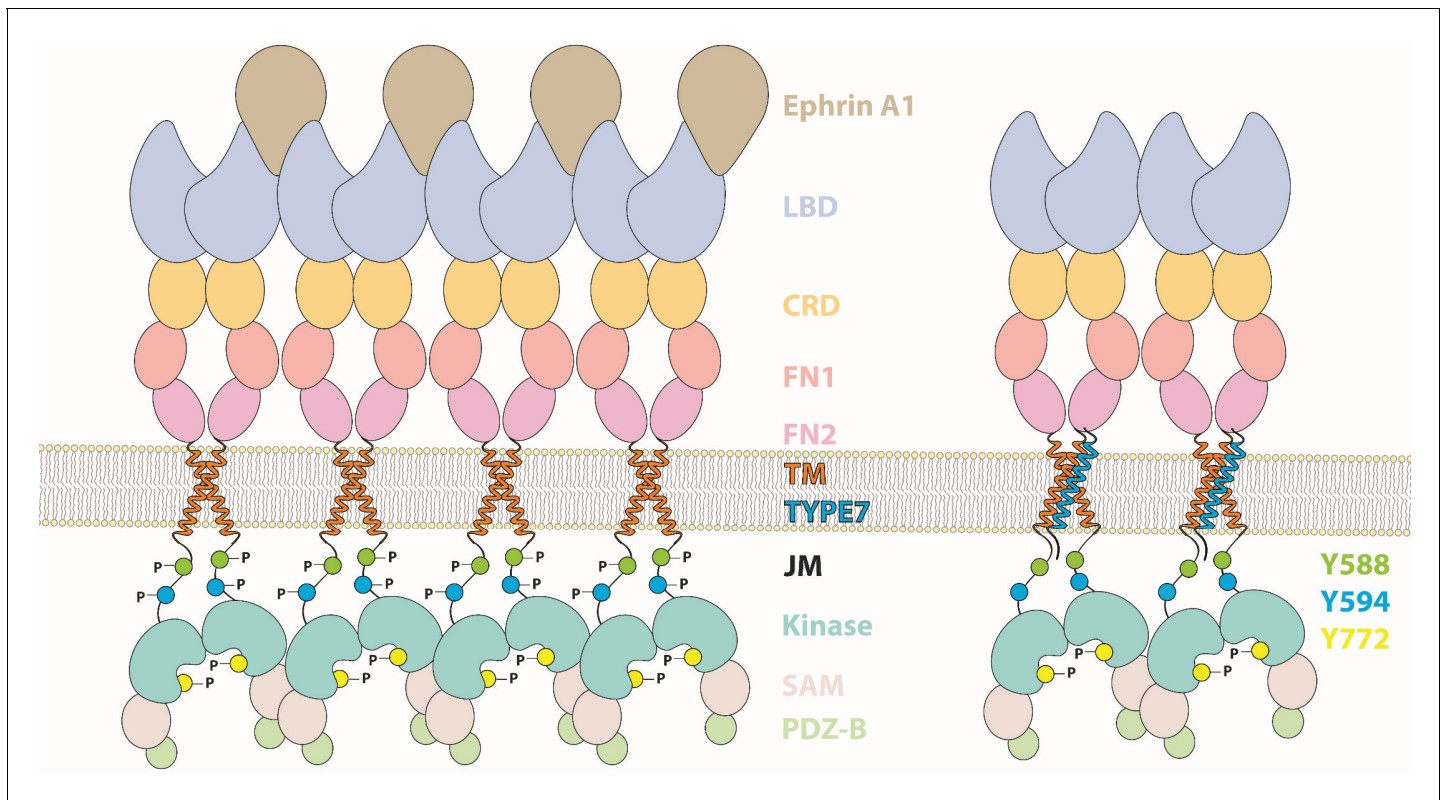

**Figure 5.** Cartoon depicting the different domains forming EphA2, which compares the activation mechanism of ephrinA1 (*left*) with the proposed TYPE7 mechanism (*right*), where the JMS is not phosphorylated and EphA2 assembles into smaller oligomers. Figure is not to scale.
DOI: https://doi.org/10.7554/eLife.36645.018

clusters might be needed instead for regulation or recycling, as a means to control the duration and intensity of EphA2 signaling (*Boissier et al., 2013*).

It has been previously shown that TM peptides can modulate other membrane receptors. However, previous efforts typically involved expressing hydrophobic peptides in cells (*Talbert-Slagle et al., 2009*; *Heim et al., 2015*), or delivering peptides solubilized using detergents and/or organic solvents (*Stone and Deber, 2017*; *Shandler et al., 2011*; *Lee et al., 2014*; *Arpel et al., 2014*), which can be deleterious to cells, and incompatible with clinical applications. Our work represents a significant advance over those efforts, since TYPE7 targets cells in physiological conditions. Furthermore, the pH-dependent membrane insertion could potentially confer a means for the targeted delivery of TYPE7 to cells in acidic environments, such as tumors.

EphA2 is a promising target for therapeutics of different cancer types. Overexpression of EphA2 in cancer can promote cancer progression and malignancy, and it is often associated with ephrin downregulation (*Macrae et al., 2005*). Importantly, TYPE7 can activate EphA2 in the absence of ephrins. Furthermore, it has been proposed that EphA2 monomers are pro-tumorigenic (*Singh et al., 2015*). As TYPE7 promotes oligomerization, we hypothesize it might have an anti-tumorigenic effect. Moreover, TYPE7 inhibits cell migration without showing toxicity, making this peptide an interesting lead compound to reduce migration of cancerous cells and metastasis. Importantly, the strategy we have developed to target EphA2 can be generalized to design peptide tools to study the activation mechanism of other single-span and multi-span membrane receptors.

## Materials and methods

### Reagents and peptides

Peptides (TYPE7 and $TMJM_{563}$-EphA2) were synthesized by Thermo Fisher Scientific (Waltham, MA) at $\geq$95% purity. Peptide purity was confirmed by matrix-assisted laser desorption ionization-time-of-flight (MALDI-TOF) mass spectrometry and high performance liquid chromatography (HPLC). The matrix $\alpha$-cyano-4-hydroxycinnamic acid ($\alpha$-HCCA) and trifluoroacetic acid (TFA) were purchased from Sigma-Aldrich (St. Louis, MO). Sodium phosphate and sodium acetate buffers were also purchased from Sigma-Aldrich (St. Louis, MO). HPLC-grade water and methanol were purchased from Fisher Chemical (Waltham, MA). Succinimidyl 6-(N-(7-nitrobenz-2-oxa-1,3-diazol-4-yl)amino) hexanoate (NBD-X, SE) was purchased from AnaSpec, Inc (Fremont, California). BODIPY FL-X SE, Alexa Fluor 568 SE, and DyLight 680 maleimide were purchased from Thermo-Fisher Scientific (Waltham, MA). Anti-EphA2 polyclonal antibody (EphA2 D4A2 XP), phospho-EphA2 (Y588-D7 $\times$ 2L), phospho-EphA2 (Y594), phospho-EphA2 (Y772), phospho-EphA2 (Y897-D9A1) and EphA2 (8B6) mouse antibody, Akt pan, phospho-Akt T308 and phospho-Akt S473 were purchased from Cell Signaling Technology (Danvers, MA). The anti-$\beta$-actin antibody was purchased from Abcam (Cambridge, MA).

### MALDI-TOF

Peptides were added to a saturated solution of $\alpha$-HCCA in 70% methanol with 0.05% TFA. The resulting solution was dried onto the MSP AnchorChip target plate (Bruker, Billerica, MA) using the dried droplet method (*Karas and Hillenkamp, 1988*). The Bruker Microflex MALDI-TOF mass spectrometer was calibrated with the Bruker Peptide Calibration Standard II (Billerica, MA). Mass spectra were analyzed using FlexAnalysis software (Bruker, Billerica, MA).

### HPLC

To check purity, analytes (peptides, peptide-dye conjugates) were dissolved in methanol and injected into a semi-preparative Agilent Zorbax 300 SB-C18 column on an Agilent 1200 series HPLC system (Santa Clara, CA). The gradient from solvent A (water +0.05% TFA) to solvent B (methanol +0.05% TFA) was 50 min from 5% B to 100% B. Peptides typically eluted near 95 – 100% B.

### Peptide conjugation

TYPE7 was labeled at the N-terminus with NBD-X SE, DyLight 680 maleimide, and BODIPY FL-X SE. Unreacted dye was removed using HPLC or gel filtration through a PD-10 column (Life Technologies, Waltham, Massachusetts), and MALDI-TOF was used to determine that a single dye molecule was bound per peptide molecule with $\alpha$-HCCA matrix.

## Liposome preparation

Lipids were purchased from Avanti Polar Lipids, Alabaster, AL. POPC (1-palmitoyl-2-oleoyl-sn-glycero-3-phosphocholine) and 22:1-PC (1,2-dierucoyl-sn-glycero-3-phosphocholine) stocks were prepared in chloroform. Aliquots of lipids were dried under a steady stream of argon gas and then placed in a vacuum overnight. The lipid films were resuspended with 10 mM sodium phosphate buffer (pH 7.9) and were then extruded with a Mini-Extruder (Avanti Polar Lipids, Alabaster, AL) through a 100 nm pore size membrane (Whatman, United Kingdom) to form large unilamellar vesicles (LUVs).

## Circular dichroism (CD)

The sample was prepared by incubation of TYPE7 with POPC LUVs, for a lipid to peptide molar ratio of 200:1. To reach the desired experimental pH, the pH of the samples was adjusted with the addition of either 100 mM sodium phosphate pH 8 or 100 mM sodium acetate pH 4. CD spectra were recorded on a Jasco J-815 spectropolarimeter at room temperature. For the solubility study, peptide samples were prepared in either PBS (pH 7.4) or 10 mM $NaP_i$ pH 8 with a final concentration of 5 µM or 50 µM. The appropriate buffer backgrounds were subtracted.

## $pH_{50}$ determination assay

$TMJM_{563}$-EphA2 and GWALP23 stocks were prepared in trifluoroethanol. Dried films of POPC, POPC:$TMJM_{563}$-EphA2 (molar ratio of 500:1), and POPC:GWALP23 (molar ratio of 500:1) were resuspended in 1 mM $NaP_i$ pH 8. The POPC liposomes and proteo-liposomes were prepared via extrusion using a Mini Extruder to form ~100 nm large unilamellar vesicles. Lyophilized TYPE7 conjugated with NBD-X FL was also rehydrated with 1 mM $NaP_i$ pH 8 and was incubated with the liposomes and proteo-liposomes with a final concentration of 0.2 µM. The POPC:TYPE7 molar ratio was 2000:1. For the titrations, a series of 100 mM buffers (sodium acetate and sodium phosphate) were used to achieve the desired pH, while keeping the total buffer concentration constant. The final pH of each individual well was measured. Fluorescence spectra were recorded at 25°C with excitation at 470 nm and an emission range of 520 – 600 nm using a Cytation five imaging plate reader (Biotek Instruments, Winooski, VT). Appropriate lipid blanks were prepared at the lowest and highest pH. The specific blanks were averaged and subtracted accordingly. Data were analyzed by calculating the center of mass (CM) of the fluorescence spectrum using the following equation:

$$CM = \sum_{1}^{n} I_i \ \lambda_i / \sum_{1}^{n} I_i \qquad (1)$$

where $Ii$ is the fluorescence intensity measured at a wavelength $\lambda i$. (**Barrera et al., 2002**; **Royer and Scarlata, 2008**). The fluorescence center of mass (F) values at different pH were fitted to determine the $pH_{50}$, using **Equation 2**:

$$F = \left(F_A \ + F_B \ 10^{m(pH-pH_{50})} \ \right) / \left(1 + 10^{m(pH-pH_{50})}\right) \qquad (2)$$

where $F_A$ is the acidic baseline, $F_B$ is the basic baseline, m is the slope of the transition, and $pH_{50}$ is the midpoint of the curve.

## Oriented circular dichroism (OCD)

Stocks of POPC, TYPE7 and $TMJM_{563}$-EphA2 were prepared in chloroform, methanol and TFE, respectively. Appropriate aliquots of lipid and peptide (50:1 lipid to peptide molar ratio) were first dried with argon gas and then placed under vacuum overnight. The lipid-peptide film was resuspended with methanol and spread on two circular quartz slides (Hellma Analytics, Germany). To ensure complete methanol evaporation, the slides were placed in a vacuum for 24 hr. After allowing the solvent to evaporate, the samples were hydrated with 150 µL of 100 mM sodium acetate buffer pH 4 – 5 overnight in 96% relative humidity, to obtain supported bilayers. The hydrated slides were assembled into the OCD cell, which had its inner cavity filled with saturated $K_2SO_4$ to keep the samples humidified. The OCD spectra were averaged for eight different rotations at 45° angles of the cell and recorded on a Jasco J-815 spectropolarimeter at room temperature. Appropriate lipid backgrounds were subtracted.

## Partition coefficient determination

Lyophilized samples of TYPE7-NBD were rehydrated in 10 mM NaPi (pH 8) at a final concentration of 0.8 µM and incubated with increasing concentrations of POPC LUVs. Emission spectra were recorded on a BioTek Cytation5 Cell Imaging Multi-Mode Reader. Three titration curves were averaged, and the resulting fluorescence intensity at 540 nm was plotted against the concentration of POPC in molar units. Fluorescence data (F) were fitted with OriginLab using:

$$F = F_0 + \Delta F \times (K_p x)/(55.3 + K_p x) \tag{3}$$

where $F_0$ is the initial fluorescence intensity, $\Delta F$ is the change in fluorescence intensity, $x$ is the lipid concentration, and 55.3 is the molar concentration of water. *Equation 3* was used to determine the partition coefficient, $K_p$, defined as the ratio of concentrations of a compound in a mixture of two phases.

## Calcein leakage assay

POPC LUVs were prepared as described above, but the dried POPC lipid film was rehydrated with 50 mM calcein in 10 mM HEPES and 50 mM EDTA (pH 8). Free calcein was removed by gel filtration through a PD-10 column. TYPE7 was added to the calcein/LUVs suspensions at different concentrations to achieve final peptide:lipid molar ratios of 0.0025 – 0.5% and incubated for 30 min at room temperature. The calcein leakage was tracked by measuring fluorescence using a Synergy two microplate reader (BioTek, Winooski, VT) at an excitation wavelength of 485 nm and an emission wavelength of 528 nm. Complete calcein release was reached by adding 20% Triton X-100, and melittin was used as a control for a leakage-inducing peptide.

## Cell culture

H358, A375, DU-145 and COS-7 cells from ATCC (Manassas, VA) were cultured in a humidified incubator under 5% $CO_2$ in RPMI (H358), DMEM (A375) and alpha-MEM (COS-7) media (Invitrogen, Carlsbad, CA) supplemented with 10% fetal bovine serum, 50 U/mL penicillin and 50 µg/ml streptomycin. Cells were incubated overnight in serum free medium in presence or absence of TYPE7 and treated the next day with recombinant IgG1 Fc (R and D Systems, Minneapolis, MN) as a control or 0.5 µg/mL of recombinant mouse EphrinA1-Fc chimera (EA1) (R and D Systems, Minneapolis, MN) for 5 or 10 min. Cell line authentication and mycoplasma-free certification was performed by ATCC for all cell lines.

## Cell proliferation assay (MTS)

H358 cell viability was measured using the CellTiter 96 Aqueous One Solution (Promega, Madison, WI) according to the manufacturer's protocol, which uses the reagent MTS (3-(4,5-dimethylthiazol-2-yl)−5-(3-carboxymethoxyphenyl)−2-(4-sulfophenyl)−2H-tetrazolium, inner salt). Briefly, cells were seeded ($2 \times 10^3$ cells per well for proliferation and $5 \times 10^4$ for toxicity) 2 days prior the experiments in a 96 well plate, and exposed to vehicle or TYPE7 at different concentrations (0.5 µM, 1 µM and 2 µM) and 3 µg/mL of Fc or EA1 and incubated 48 hr (toxicity) or 24 hr (proliferation). The MTS assay was performed in 100 µL of DMEM phenol red free medium (Invitrogen, Carlsbad, CA) in each well and 20 µL of the CellTiter solution was added to the samples, then the plate was placed in the 37°C incubator with 5% $CO_2$ until it reached the desired color. The absorbance at 490 nm was measured in a plate reader (Synergy 2, Biotek). The results are representative of three independent experiments, performed in triplicate. Cell proliferation was expressed as the percentage of vehicle control.

## Co-localization analysis

A375 cells were plated at a seeding density of $1 \times 10^4$ cells per well in a glass-bottom 8-well slide (Ibidi, Munich, Germany) coated with 50 µg/mL rat tail collagen I (Gibco, Waltham, MA). Cells were serum starved ON. In order to block the slide surface, samples were pre-treated with DMEM containing 2 µM unlabeled TYPE7 for 1 hr at 37°C. Samples were then treated with 0.5 µg/mL EphrinA1-Fc (R and D Systems, Minneapolis, MN) and/or with 0.2 µM of TYPE7-Alexa 568 in PBS containing 1 mM $MgCl_2$ and 100 microM $CaCl_2$ ($PBS^{++}$) for 5 min at room temperature followed by a 2 min wash with $PBS^{++}$ and immediately fixed in 4% PFA. After blocking and permeabilizing, samples were

incubated with rabbit anti-EphA2 primary antibody followed by secondary antibody labelling with goat-anti rabbit IgG Alexa488 (Invitrogen Carlsbad, CA).

Cells were imaged on a confocal laser scanning microscope (Zeiss LSM 710) with 63x and 100x objectives using Zen2 blue edition software. The Pearson correlation coefficient, $r$, was determined using the ImageJ Co-localization Threshold plugin. The $r$ value can range from $-1$ for perfect exclusion to $+1$ for perfect co-localization, and 0 corresponds to random localization. We calculated $r$ for whole images to reduce biases associated to selecting ROIs. However, we expect $r$ to be higher at the plasma membrane, since a population of EphA2 was internalized, while TYPE7 remained at the plasma membrane, precluding co-localization. A second factor that reduced the measured correlation was the heterogeneous expression of EphA2, since some cells have negligible receptor levels (i.e. see red cell in the lower-right corner of *Figure 2A*). Pearson correlation coefficients were compared using a Student's $t$-test assuming unequal variance in IBM SPSS Statistics Software (version 24).

## Co-precipitation

H358 cells were incubated with lysis buffer containing 150 mM NaCl, 50 mM Tris-HCl, pH 7.4, 5 mM EDTA and 1% NP-40 with protease inhibitors and phosphatase inhibitors for 30 min at 4°C. The insoluble fraction was eliminated through centrifugation at 10,000 × g for 30 min at 4°C. After the centrifugation, the lysates were incubated with anti-EphA2 antibody and protein A conjugated to Sepharose (Pierce Chemical, Rockford, IL) for 8 hr at 4°C. To quantify the total amount of protein loaded, 20 μL of the lysates was saved. Beads were washed four times with lysis buffer. Proteins were eluted in SDS-PAGE sample buffer, separated by SDS-PAGE electrophoresis, and analyzed by Western blotting of 16.5% tricine gel to detect TYPE7-DL that was precipitated with endogenous EphA2. Equal amounts of immuno-precipitate were resolved on a 10% SDS-polyacrylamide gel, and then electrophoretically transferred to 0.45 μm nitrocellulose membranes (Bio-Rad, Hercules, CA). Total cell lysates were also subjected to immunoblot. Membranes were blocked with a milk solution (150 mM NaCl, 20 mM Tris-HCl, 5% milk (w/v), 0.1% Tween (v/v), pH 7.5) and successively probed with primary (diluted 1:1000) and IR-dye-conjugated secondary antibodies (diluted 1:10,000). Immunoreactive bands and TYPE7-DL were detected using an Odyssey Infrared Scanner (Li-Cor Biosciences, Lincoln, NE).

## Protein arrays

The human Proteome Profiler Phospho-RTK Array Kit, which covers 49 different RTKs in duplicate (catalog number ARY001B), and the 43-protein Proteome Profiler Human Phospho-Kinase Array Kit (catalog number ARY003B), were purchased from R and D systems. H358 cells were starved O.N and treated with Fc, 2 μM of TYPE7 or 0.5 μg/mL of EA1 for 10 min. The assay was performed accordingly to the manufacturer protocol. Briefly, H358 cells were lysed in the provided lysis buffer with protease and phosphatase inhibitors, then incubated overnight with the nitrocellulose membranes containing the immobilized RTK tested. The membranes were then incubated with the anti-Phospho-Tyrosine-HRP detection antibody and visualized with the kit Chemi Reagent Mix.

## Structure Illumination microscopy (SIM)

After the specific treatment, cells were fixed with 4% paraformaldehyde and subsequently permeabilized with PBS$^{++}$ containing 1 mg/mL bovine serum albumin and 0.1% Triton X-100. Nonspecific binding was blocked using goat serum dilution buffer GSDB (33% goat serum, 40 mM NaPi, pH 7.4, 450 mM NaCl, and 0.6% Triton X-100). Anti-Epha2 rabbit primary and Alexa Fluor-conjugated secondary (Invitrogen, Carlsbad, CA) antibodies were diluted in GSDB and incubated for 1 hr at room temperature. Cells were visualized on a laser scanning microscope (model LSM 510; Carl Zeiss Microimaging, Thornwood, NY). Contrast and brightness settings were chosen so that all pixels were in the linear range. Images are the product of eightfold line averaging.

## Boyden chamber assay

$1 \times 10^5$ A375 or H358 cells were starved for 24 hr before the experiment, then treated in serum-free medium. Cells were seeded on the top chamber of polycarbonate 8 μm pore size membrane costar trans-well chambers (Corning Life Sciences, Corning, NY). EA1, Fc (1 μg/mL) or TYPE7 were added

to the lower chamber together with 5% FBS. Cells were allowed to migrate for 24 hr after which the cells on top of the chamber were removed with a cotton swab, and the bottom chamber was fixed with 4% PFA. After staining with eosin and hematoxylin, the cells that passed through the filter and stayed on the undersides of inserts were counted under a bight field microscope with 20x objective. Images are representative of three independent experiments, with an average of 4 images per sample condition.

## FCS cell culture and plasmids

EphA2FL (residues 1 – 971) and EphA2ΔJ (residues 1 – 558) were amplified via PCR from human EphA2 cDNA, and cloned into pEGFP-C1 plasmid. The resulting EGFP fusion genes were inserted into a LZRS-Pac retrovirus vector and then transfected into Phoenix retroviral packaging cells for retrovirus production. DU145 cells were infected with retroviral-mediated gene transfer in the presence of 6 µg/mL polybrene and selected with 1 µg/mL of puromycin. The DU145 cells with stable EphA2 expression were cultured in a collagen-coated 10 cm dish with DMEM (10% FBS). EphA2 ICD (residues 559 – 971) was amplified via PCR from human EphA2 cDNA and cloned into pEGFP-N1 vector. The c-Src membrane localization sequence was inserted before the EphA2 gene. The resulting Myr-EphA2 ICD-GFP was transfected into COS-7 (ATCC, Manassas, VA) cells using Lipofectamine2000 (Invitrogen, Carlsbad, CA). COS-7 cells were cultured in a 10 cm dish with DMEM (10% FBS). All constructs lack the PDZ domain, as described elsewhere (*Shi et al., 2017*). Experiments with plexin A4 were performed as described elsewhere (*Marita et al., 2015*).

## FCS data collection

FCS measurements were performed with a customized inverted confocal fluorescence microscopy (Eclipse Ti, Nikon) equipped with a 100x TIRF objective (NA 1.47, oil, Nikon). The 488 nm excitation laser beam was separated from a continuum white light laser (9.7 MHz) (NKT Photonics, Denmark) using a narrow-band excitation filter (488: LL01-488-12.5) (Semrock, Rochester, New York). The beam was focused onto the live cell samples sitting in an on-stage incubator by the objective. The emission light from the sample was collected through the same objective and directed passing a 520/44 nm emission filter (FF01-520/44-25) (Semrock, Rochester, New York). The photons from the emission beam were collected by a single photon avalanche diode (SPAD) detector (Micro Photon Devices, Italy) and recorded with a time-correlated single photon counting (TCSPC) module (Picoharp 300, PicoQuant). Data was processed and analyzed with a Matlab script.

Excitation laser beam at 300 nW was focused on the live cells samples at 37°C. Laser was always parked at the edge of a flat membrane area where there was only homogenous fluorescence (*Figure 4—figure supplement 1A*). Five 15 s measurements were performed on one cell and were averaged and registered as one data point. Auto-correlation was performed on the recorded time-resolved fluorescence fluctuation traces ($F(t)$) according to the following equation:

$$G(\tau) = \frac{\langle F(t+\tau)F(t)\rangle}{\langle F(t)\rangle^2} \tag{4}$$

where $\tau$ is the lag time, $G(\tau)$ is the auto-correlation function and $\langle\ \rangle$ stands for time average. The correlation of $F(t)$ rendered auto-correlation function (ACF) curve was fitted with a diffusion model shown here:

$$G(\tau) = \frac{1}{\langle N\rangle}\frac{1-F+Fe^{-\tau/\tau_T}}{1-F}\frac{1}{1+\frac{\tau}{\tau_D}} \tag{5}$$

where $N$ is average number of fluorescent particles, $\tau_D$ is the average dwell time of fluorescent particles within the detection volume, $F$ is the fraction of molecules in the triplet state, $\tau_T$ is the triplet relaxation time. The diffusion coefficient ($D$) was calculated based on $\tau_D$,

$$D = \frac{\omega_0^2}{\tau_D} \tag{6}$$

where $\omega_0$ is the waist of the laser focus. The density was calculated by dividing $N$ with the detection area that was calibrated with standard dye molecule with known diffusion coefficient.

## Statistical analysis

Unless indicated otherwise, data are reported as mean ± standard deviation (S.D), and resulted from three or more independent experiments. To evaluate differences between sample means, Student's *t*-tests or ANOVA were performed. We used IBM SPSS (version 25) and Origin 9.1 to perform *t*-tests. For each *t*-test homogeneity was checked and the correct test assuming or not assuming equal variance was applied. The same software package was used for to perform the Mann-Whitney U test to the co-precipitation data. Statistical significance was considered as $p < 0.05$. Where multiple comparisons were performed, significance was determined by *t*-tests followed by the Benjamini-Hochberg procedure using a false discovery rate of 0.05. Effect sizes in standard deviations were determined by Hedge's *g* values as calculated in Excel 2016.

# Acknowledgements

This work was supported by NIH grants R01GM120642 (FNB) and R01NS096956 and R01CA155676 (B-CW). Work by XS, SK, and AWS was supported by the National Science Foundation under Grant No. CHE-1753060. We are thankful to Dr. Joshua Bembenek (University of Tennessee), Dr. Daniel DiMaio (Yale) and Dr. Jeff Becker (University of Tennessee) for insightful comments on the manuscript. We thank Roger Koeppe II (University of Arkansas) for providing the GWALP23 peptide, and Dr. Jin Chen (Vanderbilt University) for providing reagents. We are also thankful to Nicholas Wadsworth and Alayna Cameron for helping with experimentation. The SIM and confocal experiments were conducted at the Center for Nanophase Materials Sciences (Oak Ridge National Laboratory), which is a DOE Office of Science User Facility.

# Additional information

### Funding

| Funder | Grant reference number | Author |
| --- | --- | --- |
| National Institute of General Medical Sciences | R01GM120642 | Francisco N Barrera |
| National Institute of Neurological Disorders and Stroke | R01NS096956 | Bing-Cheng Wang |
| National Cancer Institute | R01CA155676 | Bing-Cheng Wang |
| National Science Foundation | CHE-1753060 | Xiaojun Shi<br>Soyeon Kim<br>Adam W Smith |

The funders had no role in study design, data collection and interpretation, or the decision to submit the work for publication.

### Author contributions

Daiane S Alves, Conceptualization, Data curation, Formal analysis, Investigation, Methodology, Writing—original draft, Project administration, Writing—review and editing; Justin M Westerfield, Data curation, Formal analysis, Validation, Investigation, Methodology, Writing—review and editing; Xiaojun Shi, Vanessa P Nguyen, Soyeon Kim, Bing-Cheng Wang, Investigation, Writing—review and editing; Katherine M Stefanski, Data curation, Investigation, Methodology, Writing—review and editing; Kristen R Booth, Formal analysis, Investigation; Jennifer Morrell-Falvey, Resources, Investigation, Methodology; Steven M Abel, Conceptualization, Software, Formal analysis, Validation, Investigation, Methodology; Adam W Smith, Supervision, Investigation; Francisco N Barrera, Conceptualization, Resources, Data curation, Formal analysis, Supervision, Funding acquisition, Validation, Investigation, Methodology, Writing—original draft, Project administration, Writing—review and editing

### Author ORCIDs

Daiane S Alves [iD] http://orcid.org/0000-0001-9154-4748
Justin M Westerfield [iD] http://orcid.org/0000-0002-3937-5833

Xiaojun Shi https://orcid.org/0000-0002-8060-5880
Vanessa P Nguyen http://orcid.org/0000-0002-7650-0138
Katherine M Stefanski http://orcid.org/0000-0003-3007-0598
Jennifer Morrell-Falvey http://orcid.org/0000-0002-9362-7528
Steven M Abel http://orcid.org/0000-0003-0491-8647
Adam W Smith http://orcid.org/0000-0001-5216-9017
Francisco N Barrera http://orcid.org/0000-0002-5200-7891

**Decision letter and Author response**

Decision letter https://doi.org/10.7554/eLife.36645.022
Author response https://doi.org/10.7554/eLife.36645.023

## Additional files

**Supplementary files**

• Source data 1.

DOI: https://doi.org/10.7554/eLife.36645.019

• Transparent reporting form

DOI: https://doi.org/10.7554/eLife.36645.020

**Data availability**

Source data for the figures is included as Source data 1.

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
