## [Decision Letter]

Thank you for sending your article entitled "A novel pH-dependent membrane peptide that binds to EphA2 and inhibits cell migration" for peer review at *eLife*. Your article is being evaluated by three peer reviewers, and the evaluation is being overseen by a Reviewing Editor and Jonathan Cooper as the Senior Editor.

This manuscript describes a thorough investigation of a novel peptide that inhibits migration of malignant cells by its binding to the transmembrane segment (TMS) of the receptor tyrosine kinase EphA2. Thorough biophysical studies of peptide structure, membrane integration and binding are juxtaposed with a comprehensive cell biological analysis of the effects the pH-dependent TYPE2 binding to EphA2. The discovery of a peptide that can be used to activate EphA2 through an unconventional mechanism is interesting for potential therapeutic applications.

The reviewers were positively impressed by the combination of a large body of biophysical, cell biological and modeling work that you have been able to build on a difficult system. They consider that your study will represent a significant advance to the field, but a significant number of additional evidences to support your conclusions and changes to consolidate the paper are nevertheless required.

These major points are summarized below:

1) To support their hypothesis, the authors must verify that TYPE7 does increase EphA2 kinase activity, for example, by monitoring phosphorylation of a substrate (Fang et al., 2008) and/or by using the phospho-kinase antibody array in Figure 4—figure supplement 3, which contains a number of EphA2 downstream targets.

2) The authors should add more experimental evidence that TYPE7 interacts with an EphA2 TM interface but not with another interface.

3) They should show that TYPE7 does not affect the clustering of EphA2-deltaJ-GFP if the TM domain is replaced by an unrelated TM domain and that TYPE7 does not affect the clustering of Myr-EphA2-ICD-GFP if the positively charged amino acids are eliminated.

4) Some control experiments are missing: in order to conclude that TYPE7 increases pY772 and pHLIP does not, the effects of TYPE 7 and pHLIP should be compared in the same cells and preferably in the same experiment.

5) Reviewers have concerns with the mathematical model of the kinetics and wonder if the model should be kept in the manuscript. First, it is not clear whether the model is not simply custom-built to produce the observed results and has actually provided new insight. This point should be clarified. Moreover, some predictions of the model are not consistent with published data: for instance, literature does not support the presence of a substantial population of inactive EphA2 dimers in the absence of ephrin-As (see for example Singh et al., 2015 and Sabet et al., 2015; Singh et al., 2018 Communications Biology 15; and, for EphB2, Ojosnegros et al., 2017 PNAS 114:13188). According to the proposed model, ephrinA1 stimulation disrupts the EphA2 inactive dimers and promotes formation of active dimers that use a different TM interface. What are the predictions of the model in regard to the behavior the two EphA2 mutants with a disrupted TM interface and are they consistent with the data in Sharonov et al., 2014, where mutation of each TM interface affects EphA2 activity in both unstimulated and ephrinA1-stimulated cells?

Reviewer #1:

This is an impressive and thorough study of an interesting engineered peptide that builds off of the pHLIP concept. The authors are to be commended for the breadth and scope of the study, which includes a wide-range of both biophysical and cellular experiments, complemented with mathematical modeling. Relevant controls are run, including WALP and pHLIP. The paper offers several suggestions as to the mechanism through which their peptide activates EphA2, although the actual mechanism is left unresolved. I will say that I was surprised that the peptide acts to activate, rather than inhibit, the receptor. If true, the finding is very exciting and opens the door to understanding a new mechanism of activation that couples conformational changes in the TM domain to activation through oligomerization. This is an exciting mechanism that deserves publication in a high impact journal.

I have two major comments, which if adequately addressed will convince me that the paper should be published in *eLife*.

1) The authors state in reference to Figure 3—figure supplement 4D-E that there is no significant increase in EphA2 levels upon treatment with TYPE7. However, the figure does appear to indicate that there is at least some increase in expression (approximately 30%). Since we don't know if there is a relationship between expression levels of EphA2 and activation, it would be necessary to rule out the possibility that the increased (albeit mild) expression induced by TYPE7 isn't causing activation. Overexpression of other membrane proteins, such as TNF receptors, can cause ligand-free clustering and activation in cell membranes. The authors should test the effect in their cell lines of overexpression at various levels of DNA transfection to reassure the reader that this isn't affecting their core findings.

2) The use of computational modeling is commendable, but it is not clear to me that the model has actually provided new insight. More specifically, I am not convinced that the model results aren't a trivial consequence of the assumptions used to build the model. The authors should do a better job of explaining the model and making transparent how the model results differ from those that are trivial consequences of the assumptions they have started with. I will do my best to explain, and if I'm simply confused please correct the text to clarify these points:

"The model proposes that binding of saturating levels of EA1 causes a conformational switch that disfavors the inactive dimers, completely blocking the inactive interface." Then, not surprisingly, the results show an increase in the number of monomers upon ligand addition, and rapidly an increase in clusters from rapid formation of active dimers (Figure 5B). Then, adding Type 7 (Figure 5C) slows everything down. Is this not simply the consequence of the fact that, unlike the EA1 ligand, the TYPE7 effect is slowed by your choice to slow its reactivity by way of its diffusion? That is, given this difference between EA1 (instantaneous/saturating) and TYPE7 (Diffusion-slowed/concentration dependent), could you not have predicted a priori (i.e. without solving the equations) that the process would be slowed with TYPE7? I'm not saying that the mechanism for slowing isn't correct (ligand binding instantly activates pre-assembled dimers vs TYPE7 has to diffuse), I'm simply asking whether you needed a complex kinetic model to know that would be the case.

Reviewer #2:

This manuscript describes a novel peptide (TYPE7) that was designed based on the transmembrane segment of the EphA2 receptor tyrosine kinase. TYPE7 contains strategically located glutamic acid residues that make it soluble in aqueous solutions but able to insert into lipid membranes at acidic pH. Treatment of cells with TYPE7 increases phosphorylation of Y772 in the activation loop of the EphA2 kinase domain. This is attributed to the interaction of TYPE7 with the transmembrane (TM) helix and/or the juxtamembrane segment (JMS) of EphA2, which promotes EphA2 clustering and activation. The authors also developed a mathematical model of EphA2 activation based on results in the manuscript and information from the literature.

The discovery of a peptide that can be used to activate EphA2 through an unconventional mechanism is potentially interesting and could have therapeutic applications. However, additional experimental evidence is needed to support the conclusions presented and the mathematical model of EphA2 clustering.

The authors found that TYPE7 increases EphA2 phosphorylation on Y772 but not Y588 and Y594. They hypothesize that this is due to a new mechanism of EphA2 activation that does not involve Y588 and Y594 phosphorylation. To support this hypothesis, the authors should verify that TYPE7 does in fact increase EphA2 kinase activity (for example, by monitoring phosphorylation of a substrate (Fang et al., 2008) and/or by using the phospho-kinase antibody array in Figure 4—figure supplement 3, which contains a number of EphA2 downstream targets). Of note, pY588 and pY594 are major EphA2 autophosphorylation sites, and thus they would be expected to be phosphorylated in the active receptor even if their phosphorylation does not play a role in receptor activation. It should also be noted that Locard‐Paulet et al., 2016, implicated pY772 in cancer cell migration/invasiveness, but this effect may not be linked to EphA2 kinase activity.

Additional controls are needed to support the proposed mechanism of EphA2 activation by TYPE7. For example, additional controls for Figure 2A, B should include clustering of an unrelated transmembrane receptor, to show that TYPE7 specifically co-clusters with EphA2. Additional controls for Figure 3A and Figure 3—figure supplement 3 should demonstrate no effects of TYPE7 on the migration of cells that do not express EphA2, to exclude non-specific effects of TYPE7 on cell migration. Furthermore, the experiments with the pHLIP peptide in Figure 3—figure supplements 2 and 3 may not be sufficient as controls if they were performed at neutral pH, since under these conditions TYPE7 but not pHLIP would insert into the membrane of cells expressing EphA2.

The authors propose that TYPE7 interacts with an EphA2 TM interface observed by NMR (hypothesized to mediate dimerization of inactive EphA2 in cells) but not another interface (hypothesized to mediate dimerization of active EphA2 in cells). However, additional experimental evidence should be included to support this model. For example Sharonov et al., 2014, reports mutations that selectively destabilize each TM interface. If the mechanism presented by the authors is correct, TYPE7 should not associate with an EphA2 mutant where the interface hypothesized to mediate formation of inactive dimers and to interact with TYPE7 is mutated.

The authors propose that TYPE7 affects EphA2 clustering by interacting with both the TM domain and the positively charged portion at the N-terminus of the JMS segment. To support this idea, the authors should show that (i) TYPE7 does not affect the clustering of EphA2-deltaJ-GFP if the TM domain is replaced by an unrelated TM domain and (ii) TYPE7 does not affect the clustering of Myr-EphA2-ICD-GFP if the positively charged amino acids are eliminated. Could an alternative explanation of the effects observed in Figure 4 be that insertion of TYPE7 changes the properties of membranes, slowing down lateral diffusion of proteins?

Different types of experiments and their controls should be carried out in the same cell line, with possible verification of some results in other cell lines. For example, the controls in Figure 3—figure supplement 2 with the pHLIP peptide should be performed in the same cell line used to examine the effects of TYPE7 (i.e. A375). There is also some concern about the use of H358 cells, in which EphA2 activation by ephrinA1 does not seem to have robust signaling effects (see Figures 3—figure supplement 2 and Figure 4—figure supplement 3).

The authors present a detailed mathematical model of EphA2 clustering. However, some of the predictions of the model do not seem to be consistent with published data. For example, evidence in the literature does not does not support the presence of a substantial population of inactive EphA2 dimers in the absence of ephrin-As (see for example Singh et al., 2015 and Sabet et al., 2015; Singh et al., 2018, Communications Biology 15; and, for EphB2, Ojosnegros et al., 2017 PNAS 114:13188). According to the proposed model, ephrinA1 stimulation disrupts the EphA2 inactive dimers and promotes formation of active dimers that use a different TM interface. What are the predictions of the model in regard to the behavior the two EphA2 mutants with a disrupted TM interface? Are these predictions consistent with the previous data in Sharonov et al., 2014, where mutation of each TM interface affected EphA2 activity in both unstimulated and ephrinA1-stimulated cells?

Reviewer #3:

This manuscript by Alves et al. describes a thorough investigation of a novel peptide that inhibits cell migration of malignant cells by its binding to the transmembrane segment (TMS) of the receptor tyrosine kinase EphA2. The peptide can be delivered in a soluble form, then integrates into the membrane upon lowering the pH, and eventually binds to EphA2 at its TMS and the adjacent juxtamembrane segment (JMS). The authors also show that one of three phosphorylation sites in the adjacent intracellular regulatory domain of EphA2 is phosphorylated upon binding of the designed peptide TYPE7, which induces dimerization of the receptor, but no further clustering that happens only upon Ephrin A1 ligand binding and phosphorylation of all three sites. Despite its stimulation of phosphorylation of only the Y772 single site, TYPE2 is as active as Ephrin A1 in inhibiting cell migration and therefore may have future therapeutic utility.

The study is overall very comprehensive and makes a convincing case for the proposed mechanism. Thorough biophysical studies of peptide structure, membrane integration and binding are juxtaposed with a comprehensive cell biological analysis of the effects the pH-dependent TYPE2 binding to EphA2.

The comments below are suggestions on how the authors could and should further improve an already very good and carefully written manuscript.

Figure 1: A) please align the sequences for easier comparison. D) There are too many binding curves superimposed on each other so that they cannot be individually distinguished. I suggest to group them in different panels or display some of them with a shifted pH scale. The bundled overlays could still be shown in a supplementary figure to make the point of the good superposition. E) In the cartoon to the right, I am not sure what the evidence is that TYPE2 binds to a dimer of EphA2. Wouldn't it be more realistic to show TYPE2 binding to a monomer displacing the other monomer of EphA2?

OCD experiments described in the last paragraph of the subsection “Sharonov et al., 2014”: Is the dashed red line really closer to the theoretical curve for a TM helix? The data look a little ambiguous to me. Are these data really representing the pH 5 situation? Reading the relevant Materials and methods section, it appears that peptide and lipids were first dried and then rehydrated to 96% relative humidity. The pH may actually be undefined in this situation and membranes at 96% RH are not the same as in bulk water or buffer and this may drive the peptide deeper into the membrane and thereby change its orientation relative to fully hydrated bilayers. This should be acknowledged. If the authors think this is not a concern, they should present comparative data at pH 5 and pH 7.

Subsection “TYPE7 shows no toxicity and binds to cells in a pH-dependent manner”: define MTS

Subsection “TYPE7 interacts with EphA2”, first paragraph: What is the evidence in Figure 1D that a fraction of TYPE7 is already in the TM state at pH 7? – It may also be helpful to state somewhere in this paragraph that TM-EphA2 was in 4-fold molar excess over TYPT7 in these experiments (according to the Materials and methods section).

The sentence “Incubation with EA1 recapitulates EphA2 trans-activation by membrane clusters of ephrinA1”, needs a literature reference.

There seems to be some confusion about the definition of the JMS and other parts (kinase domain) of the ICD. In the first paragraph of the subsection “TYPE7 interacts with EphA2”, JMS appears to be defined as the few residues adjacent to the TMS that directly abut the membrane surface, which is a common definition in many other single-span membrane proteins. However, in the second paragraph of the subsection “TYPE7 inhibits cell migration by specific EphA2 phosphorylation at Y772”, the authors seem to include many more residues including all three phosphorylation sites that are as far as 30 residues away from the TMS in their definition of JMS. Please make definitions consistent.

Figure 4: A) Is too small to see clusters vs. no clusters that authors want to highlight. B) Does the blue FCS curve possibly represent more than one diffusing component? Its shape is clearly different from the other two. E) It is true that TYPE7 binding decreased diffusion of the myristylated receptor like it did with the native receptor (subsection “TYPE7 promotes limited self-assembly of EphA2”, last paragraph), but all diffusion coefficients are much higher in the lipid-anchored compared to the protein-anchored cases. I.e. the scale is different in E vs. C and D. Nothing wrong with this, but it should be mentioned.

Mathematical model: Does a stochastic model, which this is, have a real time scale like the minutes shown in all modeling figures, or does it not have arbitrary time steps, and therefore, should the time scale in these figures not be better called arbitrary? Of course, this does not take away from the modeling, but it does not tie the figures directly to an actual experimental time scale that may be quite different.

The model as depicted in Supplementary Figure 11A also needs more explanation. There seem to be 3 different types of k(I),(on and off) and 2 different types k(A),(on and off) as shown in the different panels. Or are they all constrained to the same respective values in the model? So, how many free fit parameters are there in the model? Clarification of this point is important to better understand the main text. The Supplementary Figure 11A legend will also benefit from a more detailed panel-by-panel explanation.

---

## [Author Response]

The reviewers were positively impressed by the combination of a large body of biophysical, cell biological and modeling work that you have been able to build on a difficult system. They consider that your study will represent a significant advance to the field, but a significant number of additional evidences to support your conclusions and changes to consolidate the paper are nevertheless required.These major points are summarized below:1) To support their hypothesis, the authors must verify that TYPE7 does increase EphA2 kinase activity, for example, by monitoring phosphorylation of a substrate (Fang et al., 2008) and/or by using the phospho-kinase antibody array in Figure 4—figure supplement 3, which contains a number of EphA2 downstream targets.

To address the request of monitoring phosphorylation of an EphA2 substrate, we have studied the effect of TYPE7 on the phosphorylation of Akt. Akt is an important downstream target of EphA2, and Akt phosphorylation promotes cell migration. We provide new figure panels (Figure 3F-H) that show that TYPE7 causes an Akt de-phosphorylation that is comparable to the effect of ephrinA1-Fc (EA1), a ligand of EphA2. Akt de-phosphorylation is a well-stablished mechanism to reduce cell migration. Thus, this new finding adds on the TYPE7 mechanistic information we had already collected. We were prompted to study Akt after re-evaluating the kinase array, where we observed that TYPE7 promoted de-phosphorylation of Akt (included now as Figure 4—figure supplement 12C), now mentioned in the last paragraph of the subsection “TYPE7 promotes limited self-assembly of EphA2”. To verify the results of the kinase array, we decided to perform Western blots for two activation sites of Akt (S473 and T308), confirming the effect of TYPE7 on this kinase crucial for cell migration. The identification of the effect of TYPE7 on a crucial EphA2 downstream target is a significant addition to the manuscript.

We originally proposed to study Rac1, a separate downstream target of EphA2. We performed those experiments, and we did not detect an effect of TYPE7 on Rac1 phosphorylation. This result is not surprising, since Rac1 activation occurs when EphA2 is phosphorylated at Y735, a residue which we have not evaluated in our manuscript for lack of commercially available phospho-specific antibody.

2) They should show that TYPE7 does not affect the clustering of EphA2-deltaJ-GFP if the TM domain is replaced by an unrelated TM domain and that TYPE7 does not affect the clustering of Myr-EphA2-ICD-GFP if the positively charged amino acids are eliminated.

New FCS experiments have been performed in live cells, where the diffusion coefficient of Plexin A4 receptor has been determined in control conditions and in the presence of TYPE7. The new figure (Figure 4—figure supplement 11) shows that TYPE7 does not affect the lateral diffusion of Plexin A4-GFP. A new paragraph discussing these results has been added (subsection “TYPE7 promotes limited self-assembly of EphA2”, third paragraph). This result suggests that TYPE7 does not affect the membrane diffusion of single-pass membrane receptors in a non-specific manner, and that the effect of TYPE7 on EphA2 diffusion results from direct binding.

3) Some control experiments are missing: in order to conclude that TYPE7 increases pY772 and pHLIP does not, the effects of TYPE 7 and pHLIP should be compared in the same cells and preferably in the same experiment.

Tthe manuscript already contained on the original submission data performed with pHLIP and TYPE7, carried at the same time and with the same cell line, H358 (Figure 3—figure supplement 6C). We have performed additional experiments where we study the effect on Y772 phosphorylation of TYPE7 and pHLIP. These experiments are again performed at the same time, but now in A375 cells. Additionally, experiments with pHLIP performed at acidic pH (pH 4.2), where pHLIP is expected to be fully in the transmembrane configuration, showed no effect (Figure 3—figure supplement 6B). In the new data, TYPE7 also increases the phosphorylation of EphA2 at Y772, although to a lower extent than in H358 cells. We also include an additional figure (Figure 3—figure supplement 5), where cell migration was studied in H358 cells (previous migration data, Figure 3A, was carried out in A375 cells). TYPE7 reduces cell migration in both cell lines. However, the effect seems even larger in H358 cells, which seems to agree with the larger increase in Y772 phosphorylation in this cell line compared to A375.

4) Reviewers have concerns with the mathematical model of the kinetics and wonder if the model should be kept in the manuscript. First, it is not clear whether the model is not simply custom-built to produce the observed results and has actually provided new insight. This point should be clarified. Moreover, some predictions of the model are not consistent with published data: for instance, literature does not support the presence of a substantial population of inactive EphA2 dimers in the absence of ephrin-As (see for example Singh et al., 2015 and Sabet et al., 2015; Singh et al., 2018 Communications Biology 15; and, for EphB2, Ojosnegros et al., 2017 PNAS 114:13188). According to the proposed model, ephrinA1 stimulation disrupts the EphA2 inactive dimers and promotes formation of active dimers that use a different TM interface. What are the predictions of the model in regard to the behavior the two EphA2 mutants with a disrupted TM interface and are they consistent with the data in Sharonov et al., 2014, where mutation of each TM interface affects EphA2 activity in both unstimulated and ephrinA1-stimulated cells?

As a result of your suggestion, the latest version of the manuscript no longer contains the mathematical modeling sections. The concerns raised in this point, and also in the next one, are solely pertinent to the model. After removal of this non-central part of the manuscript, these concerns should have been addressed.

5) The authors should add more experimental evidence that TYPE7 interacts with an EphA2 TM interface but not with another interface.

Please see response to point 4.

Reviewer #1:[…] I have two major comments, which if adequately addressed will convince me that the paper should be published in eLife.1) The authors state in reference to Figure 3—figure supplement 4D-E that there is no significant increase in EphA2 levels upon treatment with TYPE7. However, the figure does appear to indicate that there is at least some increase in expression (approximately 30%). Since we don't know if there is a relationship between expression levels of EphA2 and activation, it would be necessary to rule out the possibility that the increased (albeit mild) expression induced by TYPE7 isn't causing activation. Overexpression of other membrane proteins, such as TNF receptors, can cause ligand-free clustering and activation in cell membranes. The authors should test the effect in their cell lines of overexpression at various levels of DNA transfection to reassure the reader that this isn't affecting their core findings.

We have included additional FCS data that confirms the previous Western blot EphA2 quantification. These results show that TYPE7 does not change the plasma membrane levels of EphA2FL-GFP or EphA2DJ-GFP (Figure 4—figure supplement 10C). We now mention this finding in the second paragraph of the subsection “TYPE7 promotes limited self-assembly of EphA2”.

2) The use of computational modeling is commendable, but it is not clear to me that the model has actually provided new insight. More specifically, I am not convinced that the model results aren't a trivial consequence of the assumptions used to build the model. The authors should do a better job of explaining the model and making transparent how the model results differ from those that are trivial consequences of the assumptions they have started with. I will do my best to explain, and if I'm simply confused please correct the text to clarify these points:"The model proposes that binding of saturating levels of EA1 causes a conformational switch that disfavors the inactive dimers, completely blocking the inactive interface." Then, not surprisingly, the results show an increase in the number of monomers upon ligand addition, and rapidly an increase in clusters from rapid formation of active dimers (Figure 5B). Then, adding Type 7 (Figure 5C) slows everything down. Is this not simply the consequence of the fact that, unlike the EA1 ligand, the TYPE7 effect is slowed by your choice to slow its reactivity by way of its diffusion? That is, given this difference between EA1 (instantaneous/saturating) and TYPE7 (Diffusion-slowed/concentration dependent), could you not have predicted a priori (i.e. without solving the equations) that the process would be slowed with TYPE7? I'm not saying that the mechanism for slowing isn't correct (ligand binding instantly activates pre-assembled dimers vs TYPE7 has to diffuse), I'm simply asking whether you needed a complex kinetic model to know that would be the case.

To address this concern, the mathematical model is no longer in the manuscript.

Reviewer #2:[…] The discovery of a peptide that can be used to activate EphA2 through an unconventional mechanism is potentially interesting and could have therapeutic applications. However, additional experimental evidence is needed to support the conclusions presented and the mathematical model of EphA2 clustering.The authors found that TYPE7 increases EphA2 phosphorylation on Y772 but not Y588 and Y594. They hypothesize that this is due to a new mechanism of EphA2 activation that does not involve Y588 and Y594 phosphorylation. To support this hypothesis, the authors should verify that TYPE7 does in fact increase EphA2 kinase activity (for example, by monitoring phosphorylation of a substrate (Fang et al., 2008) and/or by using the phospho-kinase antibody array in Figure 4—figure supplement 3, which contains a number of EphA2 downstream targets). Of note, pY588 and pY594 are major EphA2 autophosphorylation sites, and thus they would be expected to be phosphorylated in the active receptor even if their phosphorylation does not play a role in receptor activation. It should also be noted that Locard‐Paulet et al., 2016, implicated pY772 in cancer cell migration/invasiveness, but this effect may not be linked to EphA2 kinase activity.

We thank the reviewer for raising this concern. This prompted us to study the downstream effects of TYPE7. As a result we discovered that TYPE7 causes de-phosphorylation of Akt, a protein that promotes cell migration. The effect of TYPE7 was similar to ephrinA1-Fc. These new data (Figure 3F-H) shows that TYPE7 affects the phosphorylation of a downstream target of EphA2 that is key for cell migration.

Additional controls are needed to support the proposed mechanism of EphA2 activation by TYPE7. For example, additional controls for Figure 2A, B should include clustering of an unrelated transmembrane receptor, to show that TYPE7 specifically co-clusters with EphA2.

We studied if TYPE7 affected the diffusive properties of Plexin A4, an unrelated single-pass receptor. The data in Figure 4—figure supplement 11 show that TYPE7 does not affect the diffusion of this receptor, which further suggests that the effect of TYPE7 on EphA2 is specific.

Additional controls for Figure 3A and Figure 3—figure supplement 3 should demonstrate no effects of TYPE7 on the migration of cells that do not express EphA2, to exclude non-specific effects of TYPE7 on cell migration.

The manuscript now contains cell migration data in two different cells lines: A375, Figure 3A; and H358, Figure 3—figure supplement 5, which reinforces our previous cell migration data. However, we think that performing experiments with migratory cells that do not express EphA2 falls outside the scope of the manuscript, which already contains extensive evidence of the specificity of the effect of TYPE7.

Furthermore, the experiments with the pHLIP peptide in Figure 3—figure supplements 2 and 3 may not be sufficient as controls if they were performed at neutral pH, since under these conditions TYPE7 but not pHLIP would insert into the membrane of cells expressing EphA2.

Figure 3—figure supplement 6 now contains experiments performed with pHLIP at acidic pH. These results show that pHLIP does not induce changes in the phosphorylation of EphA2 Y772, at neutral pH (Figure 3—figure supplement 6C) or an acidic pH (Figure 3—figure supplement 6B). These results support that data already present in Figure 3—figure supplement 7, showing that pHLIP does not alter cell migration.

The authors propose that TYPE7 interacts with an EphA2 TM interface observed by NMR (hypothesized to mediate dimerization of inactive EphA2 in cells) but not another interface (hypothesized to mediate dimerization of active EphA2 in cells). However, additional experimental evidence should be included to support this model. For example Sharonov et al., 2014, reports mutations that selectively destabilize each TM interface. If the mechanism presented by the authors is correct, TYPE7 should not associate with an EphA2 mutant where the interface hypothesized to mediate formation of inactive dimers and to interact with TYPE7 is mutated.

To address this concern, the mathematical model is no longer in the manuscript.

The authors propose that TYPE7 affects EphA2 clustering by interacting with both the TM domain and the positively charged portion at the N-terminus of the JMS segment. To support this idea, the authors should show that (i) TYPE7 does not affect the clustering of EphA2-deltaJ-GFP if the TM domain is replaced by an unrelated TM domain and (ii) TYPE7 does not affect the clustering of Myr-EphA2-ICD-GFP if the positively charged amino acids are eliminated. Could an alternative explanation of the effects observed in Figure 4 be that insertion of TYPE7 changes the properties of membranes, slowing down lateral diffusion of proteins?

We thank the reviewer for this comment, raising the intriguing possibility that TYPE7 might alter the physical properties of the membrane in such a way that slowed protein diffusion. To rule out this possibility we have performed FCS experiments with a different receptor, Plexin A4. The new data contained in Figure 4—figure supplement 11 shows that TYPE7 does not affect the diffusion coefficient of Plexin A4, which suggests that TYPE7 does not increase the microviscosity of the plasma membrane.

Different types of experiments and their controls should be carried out in the same cell line, with possible verification of some results in other cell lines. For example, the controls in Figure 3—figure supplement 2 with the pHLIP peptide should be performed in the same cell line used to examine the effects of TYPE7 (i.e. A375). There is also some concern about the use of H358 cells, in which EphA2 activation by ephrinA1 does not seem to have robust signaling effects (see Figures 3—figure supplement 2 and Figure 4—figure supplement 3).

Experiments performed at the same time with TYPE7 and pHLIP in A375 cells (Figure 3—figure supplement 6B) and H358 cells (Figure 3—figure supplement 6C) show that while TYPE7 does increase P-Y772, pHLIP does not. Additional data in H358 cells support the notion that these cells are valid to study EphA2, since EA1 robustly reduces cell migration (Figure 3—figure supplement 5), and promotes phosphorylation of EphA2 Y588 (Figure 3B, D, E), and de-phosphorylates Akt (Figure 3F-H).

The authors present a detailed mathematical model of EphA2 clustering. However, some of the predictions of the model do not seem to be consistent with published data. For example, evidence in the literature does not does not support the presence of a substantial population of inactive EphA2 dimers in the absence of ephrin-As (see for example Singh et al., 2015 and Sabet et al., 2015; Singh et al., 2018, Communications Biology 15; and, for EphB2, Ojosnegros et al., 2017 PNAS 114:13188). According to the proposed model, ephrinA1 stimulation disrupts the EphA2 inactive dimers and promotes formation of active dimers that use a different TM interface. What are the predictions of the model in regard to the behavior the two EphA2 mutants with a disrupted TM interface? Are these predictions consistent with the previous data in Sharonov et al., 2014, where mutation of each TM interface affected EphA2 activity in both unstimulated and ephrinA1-stimulated cells?

To address this concern, the mathematical model is no longer in the manuscript.

Reviewer #3:[…] The comments below are suggestions on how the authors could and should further improve an already very good and carefully written manuscript.Figure 1: A) please align the sequences for easier comparison. D) There are too many binding curves superimposed on each other so that they cannot be individually distinguished. I suggest to group them in different panels or display some of them with a shifted pH scale. The bundled overlays could still be shown in a supplementary figure to make the point of the good superposition. E) In the cartoon to the right, I am not sure what the evidence is that TYPE2 binds to a dimer of EphA2. Wouldn't it be more realistic to show TYPE2 binding to a monomer displacing the other monomer of EphA2?

A) The sequences have been aligned as suggested by the reviewer.

D) The reviewer is right in pointing out that in the original version of the figures it was hard to individually distinguish some of the overlaid curves. We have changed the shading representing the confidence intervals, and we think it is now easier to discriminate the different experiments.

E) The figure was meant to show a model of the binding between TYPE7 and TM-EphA2, without making claims about specific oligomerization state or binding stoichiometry.

OCD experiments described in the last paragraph of the subsection “Sharonov et al., 2014”: Is the dashed red line really closer to the theoretical curve for a TM helix? The data look a little ambiguous to me. Are these data really representing the pH 5 situation? Reading the relevant Materials and methods section, it appears that peptide and lipids were first dried and then rehydrated to 96% relative humidity. The pH may actually be undefined in this situation and membranes at 96% RH are not the same as in bulk water or buffer and this may drive the peptide deeper into the membrane and thereby change its orientation relative to fully hydrated bilayers. This should be acknowledged. If the authors think this is not a concern, they should present comparative data at pH 5 and pH 7.

The reviewer was rightfully confused, since there was a swap between the colors (grey and black) of the two theoretical lines in the figure legend (it was correct in the main text). We thank the reviewer for spotting this error. The text is fixed in the Figure 1 legend. We have updated the Materials and methods section to clarify the point of the 96% relative humidity (subsection “Oriented circular dichroism (OCD)”). During the overnight incubation the dry sample on the slide was first fully hydrated with abundant acidic buffer. We expect that after overnight incubation the humidity of the lipid bilayers would be close to 100%. To avoid the previously added buffer to dry, a large amount of saturated salt was filling the camber where this step was carried out. That camber would have an atmosphere that was kept humid with the saturated salt (96% relative humidity). However, the sample would still be 100% hydrated. In this situation we expect the pH to be adequately controlled. We thank the reviewer for prompting us to clarify this important point. We are confident that the OCD protocol that we employ yields valid results. Specifically, we have carried out OCD experiments on three different peptides, and our results have yielded membrane orientations independently validated by other techniques using 100% hydration. Those are the GWALP23 peptide (Nguyen et al., 2018 Biophysical Journal), the pHLIP peptide (Scott et al., 2015 Biochemistry), and the ATRAM peptide (Nguyen et al., 2015).

Regarding the suggestion of carrying out OCD experiments at pH 7, we think this could not be a straightforward experiment. On the one hand, at this pH the peptide has little helical structure (Figure 1B), and a very small OCD signal is expected. Furthermore, performing this experiment has the additional complication that at neutral pH the membrane affinity of TYPE7 decreases (Figure 1C), resulting on a non-negligible amount of peptide not being bound to the membrane, but staying on solution. Any CD signal from this sample will not result from oriented peptide, but from TYPE7 on solution, causing an artifact. In the past this situation has make interpreting OCD data of pHLIP at neutral pH unreliable in our hands.

Subsection “TYPE7 shows no toxicity and binds to cells in a pH-dependent manner”: define MTS.

The chemical name of the MTS compound has been added to the Materials and methods subsection “Cell proliferation assay (MTS)”.

Subsection “TYPE7 interacts with EphA2”, first paragraph: What is the evidence in Figure 1D that a fraction of TYPE7 is already in the TM state at pH 7? It may also be helpful to state somewhere in this paragraph that TM-EphA2 was in 4-fold molar excess over TYPT7 in these experiments (according to the Materials and methods section).

The text said”a fraction of TYPE7 is already in the TM state at neutral pH“, which is based on Figure 1D in the presence of TM-EphA2 (orange symbols), showing that at pH 7 we are beyond the high pH plateau (peripheral state), and already within the sigmoidal transition where α helix is being formed. This has been discussed the first paragraph of the subsection “Cell proliferation assay (MTS)”. To clarify that here we are referring to the presence of α helix, not that this is the dominant species, we have changed the text, replacing “a fraction” with “a non-negligible fraction”. Additionally, we have introduced in the main text the presence of a molar excess of TM-EphA2 (TMJM546-EphA2) over TYPE7, as the reviewer suggested.

The sentence “Incubation with EA1 recapitulates EphA2 trans-activation by membrane clusters of ephrinA1”, needs a literature reference.

An appropriate reference has been added (subsection “Cell proliferation assay (MTS)”, last paragraph).

There seems to be some confusion about the definition of the JMS and other parts (kinase domain) of the ICD. In the first paragraph of the subsection “TYPE7 interacts with EphA2”, JMS appears to be defined as the few residues adjacent to the TMS that directly abut the membrane surface, which is a common definition in many other single-span membrane proteins. However, in the second paragraph of the subsection “TYPE7 inhibits cell migration by specific EphA2 phosphorylation at Y772”, the authors seem to include many more residues including all three phosphorylation sites that are as far as 30 residues away from the TMS in their definition of JMS. Please make definitions consistent.

We have clarified that TM-EphA2 contains a subset of the N-terminal region of the longer JMS (subsection “Cell proliferation assay (MTS)”, first paragraph. In fact, to further clarify this point we have decided to rename TM-EphA2 to recognize it includes JM residues. The new name for TM-EphA2 is TMJM_563_-EphA2, indicating that includes the JM sequence until residue 563. The name has been updated in the whole manuscript.

Figure 4: A) Is too small to see clusters vs. no clusters that authors want to highlight. B) Does the blue FCS curve possibly represent more than one diffusing component? Its shape is clearly different from the other two. E) It is true that TYPE7 binding decreased diffusion of the myristylated receptor like it did with the native receptor (subsection “TYPE7 promotes limited self-assembly of EphA2”, last paragraph), but all diffusion coefficients are much higher in the lipid-anchored compared to the protein-anchored cases. I.e. the scale is different in E vs. C and D. Nothing wrong with this, but it should be mentioned.

A) New insets have been included highlighting the clusters (Figure 4A, Figure 4A legend).

B) The reviewer is right in pointing out that our FCS experiments do not allow ruling out the presence of more than one diffusing component. However, the data fitted well to a single relaxation term (Equation 5), suggesting that the approximation we use, which is to compare the average values for the single component fits, is reasonable. We have introduced this important qualification in the second paragraph of the subsection “TYPE7 promotes limited self-assembly of EphA2”.

E) We now discuss the increased D value for the myristoylated construct in the last paragraph of the subsection “TYPE7 promotes limited self-assembly of EphA2”.

Mathematical model: Does a stochastic model, which this is, have a real time scale like the minutes shown in all modeling figures, or does it not have arbitrary time steps, and therefore, should the time scale in these figures not be better called arbitrary? Of course, this does not take away from the modeling, but it does not tie the figures directly to an actual experimental time scale that may be quite different.The model as depicted in Supplementary Figure 11A also needs more explanation. There seem to be 3 different types of k(I),(on and off) and 2 different types k(A),(on and off) as shown in the different panels. Or are they all constrained to the same respective values in the model? So, how many free fit parameters are there in the model? Clarification of this point is important to better understand the main text. The Supplementary Figure 11A legend will also benefit from a more detailed panel-by-panel explanation.

The comment should have been addressed after removal of the mathematical model from the manuscript.